# CEM500K, a large-scale heterogeneous unlabeled cellular electron microscopy image dataset for deep learning

**Ryan Conrad[1,2], Kedar Narayan[1,2]\***

[1]Center for Molecular Microscopy, Center for Cancer Research, National Cancer Institute, National Institutes of Health, Bethesda, United States; [2]Cancer Research Technology Program, Frederick National Laboratory for Cancer Research, Frederick, United States

**Abstract** Automated segmentation of cellular electron microscopy (EM) datasets remains a challenge. Supervised deep learning (DL) methods that rely on region-of-interest (ROI) annotations yield models that fail to generalize to unrelated datasets. Newer unsupervised DL algorithms require relevant pre-training images, however, pre-training on currently available EM datasets is computationally expensive and shows little value for unseen biological contexts, as these datasets are large and homogeneous. To address this issue, we present CEM500K, a nimble 25 GB dataset of $0.5 \times 10^6$ unique 2D cellular EM images curated from nearly 600 three-dimensional (3D) and 10,000 two-dimensional (2D) images from >100 unrelated imaging projects. We show that models pre-trained on CEM500K learn features that are biologically relevant and resilient to meaningful image augmentations. Critically, we evaluate transfer learning from these pre-trained models on six publicly available and one newly derived benchmark segmentation task and report state-of-the-art results on each. We release the CEM500K dataset, pre-trained models and curation pipeline for model building and further expansion by the EM community. Data and code are available at https://www.ebi.ac.uk/pdbe/emdb/empiar/entry/10592/ and https://git.io/JLLTz.

**\*For correspondence:**
kedar.narayan@nih.gov

**Competing interests:** The authors declare that no competing interests exist.

## Introduction

Accurate image segmentation is essential for analyzing the structure of organelles and cells in electron microscopy (EM) image datasets. Segmentation of volume EM (vEM) data has enabled researchers to address questions of fundamental biological interest, including the organization of neural circuits (*Takemura et al., 2015*; *Kasthuri et al., 2015*) and the structure of various organelles (*Vincent et al., 2017*; *Vincent et al., 2019*; *Hoffman et al., 2020*). Truly automated EM image segmentation methods hold the promise of significantly accelerating the rate of discovery by enabling researchers to extract and analyze information from their datasets without months or years of tedious manual labeling. While supervised deep learning (DL) models are effective at the segmentation of objects in natural images (e.g. of people, cars, furniture, and landscapes) (*Wang et al., 2020*; *Tao et al., 2020*; *Carion et al., 2020*; *He et al., 2020*), they require significant human oversight and correction when applied to the organelles and cellular structures captured by EM (*Lichtman et al., 2014*; *Plaza and Funke, 2018*).

Many of the limitations of supervised DL segmentation models for cellular EM data result from a lack of large and, importantly, diverse training datasets (*Goodfellow, 2016*; *Pereira et al., 2009*; *Sun et al., 2017*). Although several annotated image datasets for cell and organelle segmentation are publicly available, these often exclusively consist of images from a single experiment or tissue type, and a single imaging approach (*Guay et al., 2020*; *Žerovnik Mekuč et al., 2020*; *Casser et al., 2018*; *Perez et al., 2014*; *Berning et al., 2015*). The homogeneity of such datasets

often means that they are ineffective for training DL models to accurately segment images from unseen experiments. Instead, when confronted with new data, the norm is to extract and annotate small regions-of-interest (ROIs) from the EM image, train a model on the ROIs, and then apply the model to infer segmentations for the remaining unlabeled data (*Guay et al., 2020*; *Žerovnik Mekuč et al., 2020*; *Casser et al., 2018*; *Perez et al., 2014*; *Berning et al., 2015*; *Januszewski et al., 2018*; *Funke et al., 2019*). Often, not only are these models dataset-specialized, reducing their utility, they often fail to generalize even to parts of the same dataset that are spatially distant from the training ROIs (*Žerovnik Mekuč et al., 2020*; *Buhmann, 2019*).

There are two main methods to address this challenge. First, gathering more annotated data for model training from disparate sources could certainly improve a model's ability to generalize to unseen images, yet it is rarely feasible for typical research laboratories to generate truly novel datasets; most have expertise in a particular imaging technique, organism, or tissue type. Beyond collecting the EM data, manual segmentation is time-consuming and, unlike for natural images, difficult to crowdsource because of the extensive domain knowledge required to identify objects in novel cellular contexts. Promising work is being done in the area of citizen science as it pertains to EM data, but it is clear that there are limitations to the range of structures that can be accurately segmented by volunteers (*Spiers, 2020*; *EyeWirers et al., 2014*). Moreover, structure-specific annotations will not solve the generalization problem for all possible EM segmentation targets; for example, thousands of hours spent labeling neurites is unlikely to buy any gains for mitochondrial segmentation.

The second and less costly method is to use transfer learning. In transfer learning, a DL model is pre-trained on a general task and its parameters are reused for more specialized downstream tasks. A well-known example is to transfer parameters learned from the ImageNet classification task (*Deng et al., 2015*) to other classification or object detection tasks which have fewer training examples (*Ren et al., 2017*). Transfer learning, when relevant pre-trained parameters are available, is the default approach for extracting the best performance out of small training datasets (*Huh et al., 2016*; *Devlin et al., 2018*). While ImageNet pre-trained models are sometimes used for cellular EM segmentation tasks (*Karabağ et al., 2020*; *Devan et al., 2019*), high-level features learned from ImageNet may not be applicable to biological imaging domains (*Raghu et al., 2019*). Building a more domain-specific annotated dataset large enough for pre-training would be a significant bottleneck, and indeed, it required multiple years to annotate the $3.2 \times 10^6$ images that form the basis of ImageNet. Fortunately, recent advances in unsupervised learning algorithms have now enabled effective pre-training and transfer learning without the need for any up-front annotations; in fact, on many tested benchmarks, unsupervised pre-training leads to better transfer learning performance (*Tian et al., 2019*; *Chen et al., 2020a*; *He et al., 2019*; *Donahue and Simonyan, 2019*; *Ji et al., 2018*; *Wu et al., 2018*; *Kolesnikov, 2019*).

To provide a resource for the EM community to explore these exciting advances, we constructed an unlabeled cellular EM dataset which we call CEMraw, containing images from 101 unrelated biological projects. The image data superset, comprising 591 3D image volumes and 9,626 2D images, is collated from a collection of experiments conducted in our own laboratory as well as data from publicly available sources. After gathering this set of heterogeneous images, we create a pipeline where we first remove many nearly identical images and then filter out low-quality and low-information images. This results in a highly information-rich, relevant, and non-redundant 25 GB 2D image dataset comprising $0.5 \times 10^6$ images. As a proof of concept for its potential applications, we pre-trained a DL model on CEM500K using an unsupervised algorithm, MoCoV2 (*Chen et al., 2020b*), and evaluated the results for transfer learning on six publicly available benchmarks: CREMI Synaptic Clefts (*CREMI, 2016*), Guay (*Guay et al., 2020*), Kasthuri++ and Lucchi++ (*Casser et al., 2018*), Perez (*Perez et al., 2014*) and UroCell (*Žerovnik Mekuč et al., 2020*).and one newly derived benchmark that we introduce in this work. CEM500K pre-trained models significantly outperformed randomly initialized and ImageNet pre-trained models, as well as previous baseline results from benchmark-associated publications.

## Results

### Creation of CEM500K

In order to create an image dataset that is relevant to cellular EM and yet general enough to be applicable to a variety of biological studies and experimental approaches, we collected 2D and 3D cellular EM images from both our own experiments and publicly available sources. These included images from a variety of imaging modalities and their corresponding sample preparation protocols, resolutions reported, and cell types imaged (*Figure 1a–c*). We selected 'in-house' datasets corresponding to 251 reconstructed FIB-SEM volumes from 33 unrelated experiments and 2975 transmission EM (TEM) images from 35 additional experiments. Other data was sourced externally; as there is currently no central hub for accessing publicly available datasets, we manually searched through databases (Cell Image Library, Open Connectome Project [*Vogelstein et al., 2018*], EMPIAR [*Iudin et al., 2016*]), GitHub repositories, and publications. A complete accounting of the datasets with relevant attribution is detailed in the **Supplementary Materials**. Included in this batch of data were 340 EM image volumes (some derived from video data) from 26 experiments and 9792 2D

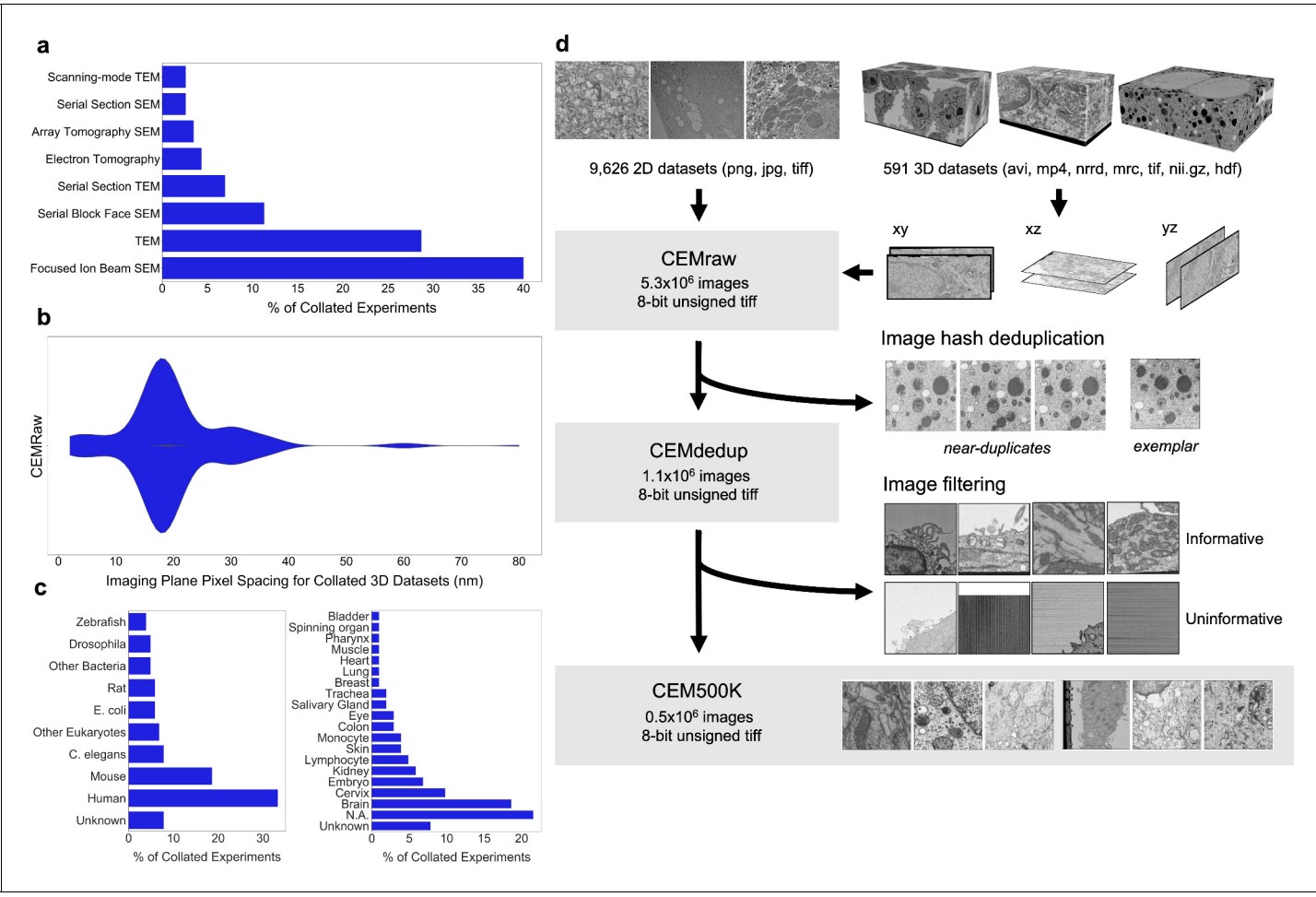

**Figure 1.** Preparation of a deep learning appropriate 2D EM image dataset rich with relevant and unique features. (**a**) Percent distribution of collated experiments grouped by imaging technique: TEM, transmission electron microscopy; SEM, scanning electron microscopy. (**b**) Distribution of imaging plane pixel spacings in nm for volumes in the 3D corpus. (**c**) Percent distribution of collated experiments by organism and tissue origin. (**d**) Schematic of our workflow: 2D electron microscopy (EM) image stacks (top left) or 3D EM image volumes sliced into 2D cross-sections (top right) were cropped into patches of 224 × 224 pixels, comprising CEMraw. Nearly identical patches excepting a single exemplar were eliminated to generate CEMdedup. Uninformative patches were culled to form CEM500K.

The online version of this article includes the following source data for figure 1:

**Source data 1.** Details of imaging technique, organism, tissue type and imaging plane pixel spacing in collated imaging experiments.

images from 14 other experiments. Among the externally gathered datasets, there were disparate file types (avi, mp4, png, tiff, jpeg, mrc, nii.gz) and pixel/voxel data types (signed and unsigned, 32-bit float, 8-bit and 16-bit integer) as well as a mixture of image volumes with isotropic or anisotropic voxels, and regular or inverted intensities. These data were standardized into 2D tiff images, or patches, of 224 $\times$ 224 8-bit unsigned pixels (see 'Materials and methods'); the resulting set of 5.3 $\times$ $10^6$ images constitutes what we term CEMraw (*Figure 1d*, top).

Within CEMraw, however, most images were redundant. Nearly identical patches existed because of the similarity between adjacent cross-sections in high-resolution 3D volumes as well as in patches cropped from uniform intensity regions like empty resin. Duplicates are not only memory and computationally inefficient, but they may also induce undesirable biases toward the most frequently sampled features in the dataset. Therefore, we aggressively removed duplicates using an automated algorithm: we calculated and compared image hashes for each patch in CEMraw and then kept a single, randomly chosen exemplar image from each group of near-duplicates (see 'Materials and methods'). As a result of this operation, we obtained an 80% decrease in the number of patches when compared to CEMraw; this 'deduplicated' subset of 1.1 $\times$ $10^6$ image patches we refer to as CEMdedup (*Figure 1d*, middle).

Deduplication ensures that each image will make a unique contribution to our dataset, but it is agnostic to the content of the image, which may or may not be relevant to downstream tasks. Upon visual inspection, it was clear that many of the images in CEMdedup contained little information useful to the segmentation of organelles or cellular structures, for example, images dominated by empty resin, background padding, or homogeneously stained interiors of nuclei or cytoplasm (*Appendix 1—figure 1a*). However, while these images were uninformative for our purposes, they also represented a wide variety of image features, making them challenging to identify with simple image statistics. Instead, we separated an arbitrary subset of 12,000 images from CEMdedup into informative and uninformative classes and trained a DL model to perform binary classification on the entire dataset. Uninformative images were characterized by poor contrast, large areas of uniform intensity, artifacts, and the presence of non-cellular objects. Detailed criteria are given in 'Materials and methods'. The classifier achieved an area under the receiver operating characteristic (AUROC) score of 0.962 on a holdout test set of 2000 images, as shown in *Appendix 1—figure 1b*, suggesting that it could reliably distinguish between the informative and uninformative image classes. Classification of the remaining unlabeled images with this model yielded 0.5 $\times$ $10^6$ patches with a visibly higher density of views containing organelles and cellular structures. We refer to this final subset of uniquely informative 2D cellular EM images as CEM500K (*Figure 1d*, bottom). Representative patches from the three datasets (CEMraw, CEMdedup, and CEM500K) are shown in *Appendix 1—figure 2*.

## Test of pre-training by CEM500K

We then decided to test CEM500K for unsupervised pre-training of a DL model, using the MoCoV2 algorithm, a relatively new and computationally efficient approach (*He et al., 2019*). The algorithm works by training a DL model to match differently augmented (e.g., cropped, rotated, zoomed in, and brightened) pairs of images. The first batch of augmented images is called the query and the batch of their differently augmented counterparts is called the key. Before matching, the encoded images in the key are added to a continuously updated queue containing tens of thousands of recently seen images (*Appendix 1—figure 3a*). To be useful for other tasks, it is assumed that the model will learn features that correspond to relevant objects within the training images. Recently, models pre-trained on ImageNet with the MoCoV2 algorithm have shown superior transfer learning performance over supervised methods when applied to a variety of tasks including segmentation (*Chen et al., 2020b*). Before we were able to evaluate the MoCoV2 algorithm on CEM500K, it was necessary to define a set of downstream tasks to quantify and compare performance. We chose six publicly available benchmark datasets: CREMI Synaptic Clefts (*CREMI, 2016*), Guay (*Guay et al., 2020*), Kasthuri++ and Lucchi++ (*Casser et al., 2018*), Perez (*Perez et al., 2014*), and UroCell (*Žerovnik Mekuč et al., 2020*). The benchmarks included a total of eight organelles or subcellular structures for segmentation (mitochondria, lysosomes, nuclei, nucleoli, canalicular channels, alpha granules, dense granules, dense granule cores, and synaptic clefts). In *Figure 2a*, we show representative images and label maps from the benchmarks. Additional information about the benchmarks,

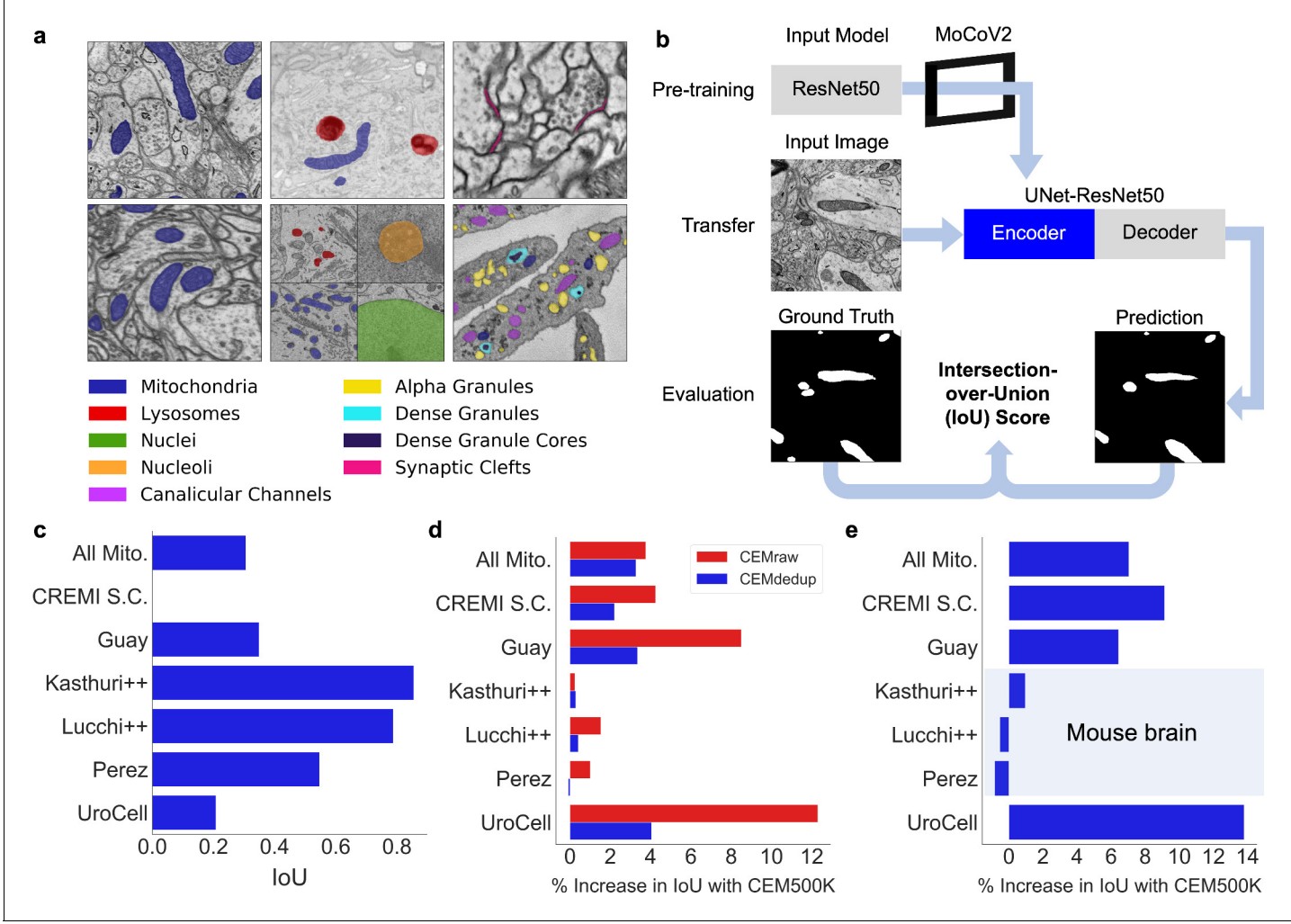

**Figure 2.** CEM500K pre-training improves the transferability of learned features. (**a**) Example images and colored label maps from each of the six publicly available benchmark datasets: clockwise from top left: Kasthuri++, UroCell, CREMI Synaptic Clefts, Guay, Perez, and Lucchi++. The All Mitochondria benchmark is a superset of these benchmarks and is not depicted. (**b**) Schematic of our pre-training, transfer, and evaluation workflow. Gray blocks denote trainable models with randomly initialized parameters; blue block denotes a model with frozen pre-trained parameters. (**c**) Baseline Intersection-over-Union (IoU) scores for each benchmark achieved by skipping MoCoV2 pre-training. Randomly initialized parameters in ResNet50 layers were transferred directly to UNet-ResNet50 and frozen during training. (**d**) Measured percent difference in IoU scores between models pre-trained on CEMraw vs. CEM500K (red) and on CEMdedup vs. CEM500K (blue). (**e**) Measured percent difference in IoU scores between a model pre-trained on CEM500K over the mouse brain (Bloss) pre-training dataset. Benchmark datasets comprised exclusively of electron microscopy (EM) images of mouse brain tissue are highlighted.

The online version of this article includes the following source data for figure 2:

**Source data 1.** IoU scores achieved with different datasets used for pre-training.

including imaging techniques and sizes of the training and test sets, is given in *Supplementary file 1*.

Performance on each benchmark was measured using the standard Intersection-over-Union (IoU) score. Considered on their own, many of these benchmark datasets are not difficult enough to expose the gap in performance between different models: they only require the segmentation of a single organelle within a test set that is often from the same image volume as the training set. At the same time, they are an accurate reflection of common use cases for deep learning in EM laboratories where the goal is to segment data from a single experiment in order to support biological, not computational, research. To address the lack of variety within the benchmark training and test sets, we derived an additional benchmark that we call All Mitochondria, which is a combination of the

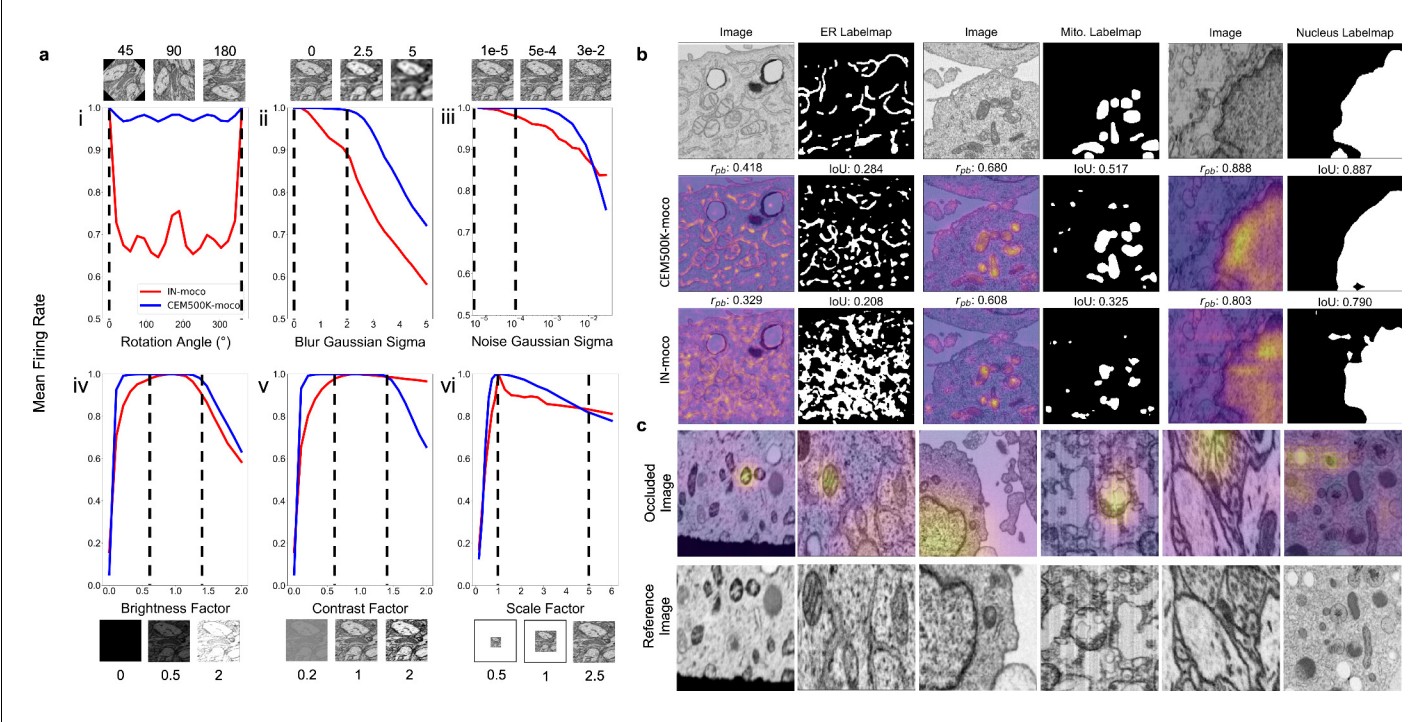

**Figure 3.** Features learned from CEM500K pre-training are more robust to image transformations and encode for semantically meaningful objects with greater selectivity. (**a**) Mean firing rates calculated between feature vectors of images distorted by (i) rotation, (ii) Gaussian blur, (iii) Gaussian noise, (iv) brightness v. contrast, (vi) scale. Dashed black lines show the range of augmentations used for CEM500K + MoCoV2 during pre-training. For transforms in the top row, the undistorted images occur at x = 0; bottom row, at x = 1. (**b**) Evaluation of features corresponding to ER (left), mitochondria (middle), and nucleus (right). For each organelle, the panels show: input image and ground truth label map (top row), heatmap of CEM500K-moco activations of the 32 filters most correlated with the organelle and CEM500K-moco binary mask created by thresholding the mean response at 0.3 (middle row), IN-moco activations and IN-moco binary mask (bottom row). Also included are Point-Biserial correlation coefficients ($r_{pb}$) values and Intersection-over-Union scores (IoUs) for each response and segmentation. All feature responses are rescaled to range [0, 1]. (**c**) Heatmap of occlusion analysis showing the region in each occluded image most important for forming a match with a corresponding reference image. All magnitudes are rescaled to range [0, 1].

training and test sets from each of the five benchmarks that contain label maps for mitochondria (Guay, Perez, UroCell, Lucchi++, and Kasthuri++; the labels for all other objects were removed). Although this benchmark is specific to a single organelle, it is challenging in that it requires a model to learn features that are general for mitochondria from image volumes generated independently and from unrelated experiments and imaging parameters.

Our overall pre-training, transfer, and evaluation workflow is shown in a schematic in *Figure 2b*. Pre-training was performed by applying the MoCoV2 algorithm to learn parameters for a ResNet50 (*He et al., 2016*) before transferring the parameters into the encoder of a U-net (*Ronneberger et al., 2015*). A detailed schematic of the UNet-ResNet50 architecture is shown in *Appendix 1—figure 3b*. For this section, once transferred, the parameters were frozen such that no updates were made during training on the benchmark tasks; this enabled us to isolate the effects of pre-training. As a simple baseline reference for calibrating later results, we started by measuring the performance of the proposed segmentation model with randomly initialized and frozen encoder parameters (i.e., we skipped the pre-training step in the workflow); the results for each benchmark are shown *Figure 2c*. Given that in our architecture, the encoder includes approximately $23 \times 10^6$ parameters and the decoder approximately $9 \times 10^6$ parameters, some 70% of the model's parameters were never updated during training. Still, some benchmarks permit strikingly good performance, with IoU scores of over 0.75 on both Lucchi++ and Kasthuri++. These results emphasize the necessity of evaluating deep learning algorithms and pre-training datasets on multiple benchmarks before drawing conclusions about their quality.

We next tested the influence of our curation pipeline on the quality of pre-trained parameters. We pre-trained models on CEMraw, CEMdedup, and CEM500K with an abbreviated training schedule (see 'Materials and methods') and compare the IoU scores achieved on the benchmarks in *Figure 2d* (the actual IoU scores are shown in *Table 1*). We observed that pre-training on CEM500K gave better or equivalent results than the CEMraw superset and CEMdedup subset for every benchmark. The average increase in performance of CEM500K over CEMraw was 4.5%, and CEM500K over CEMdedup was 2.0%, with a maximum increase of 12.3% and 4.1%, respectively, on the Uro-Cell benchmark (IoU scores increased from 0.652 and 0.699 to 0.729). These increases are significant. As a comparison, a 2% increase in model performance is similar in magnitude to what might be expected from using an ensemble of a few models (*Ju et al., 2017*). Besides these gains, curation is valuable for reducing the computational cost of using CEM500K: the final filtered subset is 90% smaller than the raw superset (25 GB compared to 250 GB). Deduplication and filtering likely contributed to the performance gain by enabling both faster convergence and the learning of more relevant feature detectors. Duplicate images consume training iterations without presumably transmitting any new information, resulting in slower learning. Uninformative images, on the other hand, may guide a model to discover discriminative features that are useless for most segmentation tasks. For example, a model must learn feature detectors that can distinguish between images of empty resin in order to succeed on the pre-training task, but those feature detectors are unlikely to help with a common task like mitochondrial segmentation. Therefore, eliminating uninformative images may reduce the learning of irrelevant details during pre-training.

We also posited that, in addition to the benefits of curation, the heterogeneity of examples in CEM500K would be essential for achieving good segmentation performance across disparate biological contexts. To test this, we considered an alternative pre-training dataset consisting exclusively of $1 \times 10^6$ images from a single large connectomics volume of mouse brain tissue (*Bloss et al., 2018*). Coming from a single volume of a highly homogeneous tissue type, images in this dataset show much less variation in cellular features than those in CEM500K (a random sampling of images is shown in *Appendix 1—figure 4*). The size of the volume and the density of its content allowed us to sparsely sample patches without the need for deduplication and filtering.

Compared to the Bloss pre-training dataset, CEMraw, CEMdedup, and CEM500K all demonstrated significantly higher performance on four of the seven benchmarks, as shown in *Figure 2e* (the actual IoU scores are shown in *Table 1*). The average increase in IoU scores from the Bloss baseline to CEM500K over these four benchmarks was 9.1%, with a maximum of 13.8% for the UroCell benchmark (increase in IoU score from 0.638 to 0.729). Tellingly, the three benchmarks on which Bloss pre-trained models performed comparably well (Kasthuri++, Lucchi++, and Perez) were the only benchmarks that exclusively contained images from mouse brain tissue, like the Bloss dataset itself. This apparent specificity for images from the same organism and tissue type may indicate that

**Table 1.** Comparison of segmentation Intersection-over-Union (IoU) results for benchmark datasets from models randomly initialized and pre-trained with MoCoV2 on the Bloss dataset, and CEMraw, CEMdedup, and CEM500K.
* denotes benchmarks that exclusively contain electron microscopy (EM) images from mouse brain tissue. The best result for each benchmark is highlighted in bold and underlined.

| Benchmark | Random Init. (No Pre-training) | Bloss et al., 2018 | CEMraw | CEMdedup | CEM500K |
|---|---|---|---|---|---|
| All Mitochondria | 0.306 | 0.694 | 0.719 | 0.722 | <u>0.745</u> |
| CREMI Synaptic Clefts | 0.000 | 0.242 | 0.254 | 0.259 | <u>0.265</u> |
| Guay | 0.349 | 0.380 | 0.372 | 0.391 | <u>0.404</u> |
| *Kasthuri++ | 0.855 | 0.907 | 0.913 | 0.913 | <u>0.915</u> |
| *Lucchi++ | 0.788 | <u>0.899</u> | 0.880 | 0.890 | 0.894 |
| *Perez | 0.547 | <u>0.874</u> | 0.854 | 0.866 | 0.869 |
| UroCell | 0.208 | 0.638 | 0.652 | 0.699 | <u>0.729</u> |
| *Average Mouse Brain | 0.730 | <u>0.893</u> | 0.883 | 0.890 | <u>0.893</u> |
| Average Other | 0.216 | 0.489 | 0.499 | 0.518 | <u>0.536</u> |

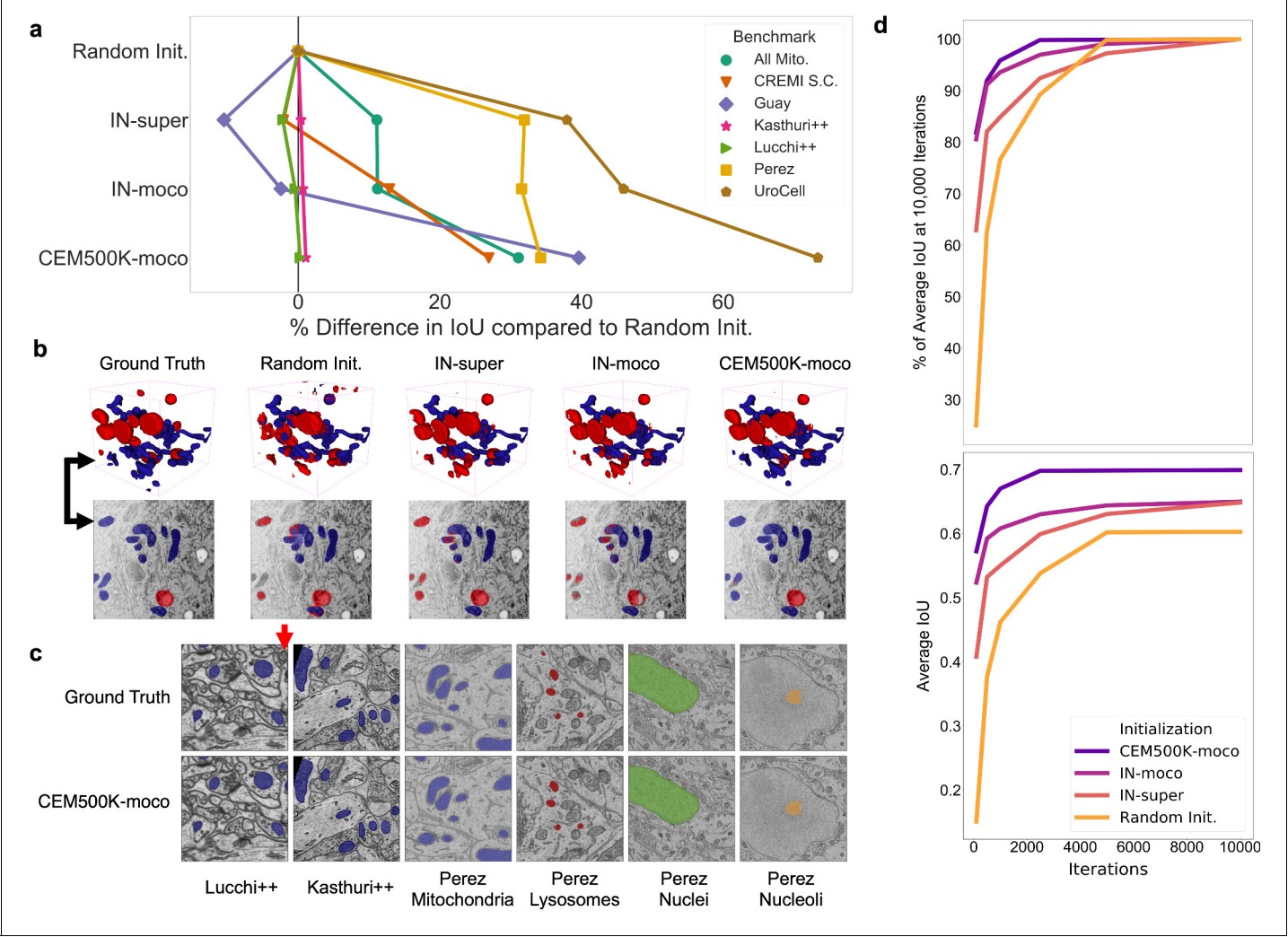

**Figure 4.** Models pre-trained on CEM500K yield superior segmentation quality and training speed on all segmentation benchmarks. (a) Plot of percent difference in segmentation performance between pre-trained models and a randomly initialized model. (b) Example segmentations on the UroCell benchmark in 3D (top) and 2D (bottom). The black arrows show the location of the same mitochondrion in 2D and in 3D. (c) Example segmentations from all 2D-only benchmark datasets. The red arrow marks a false negative in ground truth segmentation detected by the CEM500K-moco pre-trained model. (d) Top, average IoU scores as a percent of the average IoU after 10,000 training iterations, bottom, absolute average IoU scores over a range of training iteration lengths.

The online version of this article includes the following source data for figure 4:

**Source data 1.** IoU scores for different pre-training protocols.
**Source data 2.** IoU scores for different training iterations by pre-training protocol .

the models learn to represent elements of the underlying biology or tissue architecture. Alternatively, it may reflect similarities in the image acquisition and sample preparation protocols, though the plausibility of this explanation is unlikely, given that each benchmark dataset was imaged with different, albeit broadly similar, technologies (Bloss with serial section TEM; Kasthuri++ with ATUM-SEM; Lucchi++ with FIB-SEM; Perez with SBF-SEM). It is clear that pre-training on large but biological narrow datasets is insufficient for learning general-purpose features that apply equally well across a broad spectrum of contexts. To guard against potential biases, our results instead suggest that the pre-training dataset ought to include image examples from as many different tissues, organisms, sample preparation protocols, and EM techniques as possible. Furthermore, a set of diverse benchmark datasets is essential for identifying such biases when they do arise.

## CEM500K models are largely impervious to meaningful image augmentations

Having established CEM500K as the EM dataset for pre-training and transfer learning, we investigated the qualities of the model pre-trained by the MoCoV2 algorithm on CEM500K and compare it to a model pre-trained by the MoCoV2 algorithm on ImageNet (IN-moco). We note that unlike the abbreviated training used to evaluate pre-training on various subsets of cellular electron microscopy (CEM), here we trained the model for the complete schedule, and henceforth refer to the fully trained model as CEM500K-moco. In general, good DL models have neurons that are both robust to distortions and are selective for particular features (*Goodfellow et al., 2009*). In the context of EM images, for example, a good model must be able to recognize a mitochondrion as such irrespective of its orientation in space, its size, or some reasonable variation in resolution of its membrane. On the other hand, the same model must also be able to discern mitochondria, no matter how heterogeneous, from a variety of other organelles or cellular features. First, we attempted to evaluate the robustness of CEM500K-moco neurons by measuring their invariances to transformations of input images. Specifically, we considered the average activations of the 2048 neurons in the last layer of the ResNet50s' encoders, pre-trained by either CEM500k-moco or IN-moco, to input images. Broadly following the approach detailed in *Goodfellow et al., 2009*, we defined invariance based on the mean firing rates of neurons in response to distortions of their inputs. Plots showing changes in mean firing rates with respect to rotation, Gaussian blur and noise, brightness, contrast, and scale are shown in *Figure 3a*. These six transforms that we choose account for much of the variation observed experimentally in cellular EM datasets, and we expect that models in which many neurons are invariant to these differences would be better suited to cellular EM segmentation tasks.

We observed that neurons in CEM500K-moco models had consistently stronger invariance to all tested transformations (*Figure 3a*). The two exceptions were a reduction in invariance when contrast was very high and a smaller reduction when scale factors were very large (*Figure 3a,v and vi*, respectively). First, with regard to rotation, virtually all the neurons in the CEM500K-moco model were remarkably invariant to rotation compared to about 70% of the neurons in the IN-moco model, reflecting the fact that orientation matters for representing images in ImageNet but, appropriately, not for CEM500K. This rotation invariance likely arises from the range of augmentations applied during pre-training. Next, neurons in the CEM500K-moco model fire more consistently when presented with increasingly blurry and noisy images, in both cases falling off significantly later as compared to IN-moco, when, presumably, meaningful information in the images has been lost. Further, while both of the tested pre-trained models responded comparably to increasing image brightness, the CEM500K-moco model had a noticeably greater invariance to both more brightened and more darkened images. For contrast adjustments, there was a similar robustness to decreased contrast. This was indicative of the distribution of images in CEM500K, and cellular EM data more broadly: very low-contrast images are common, very high-contrast images are not. On the other hand, the gap between CEM500K-moco and IN-moco pre-trained models in the high-contrast regime not only reinforces this observation but also suggests more relevant learning by the former. CEM500K-moco neurons show an invariance to a transformation only insofar as that transformation mimics real variance in the data distribution, and the firing rate decreases when the high contrast becomes no longer plausible. Similarly, there is some evidence that the results for scale invariance follow the same logic. In CEM500K, the most common reported image pixel sampling was 15–20 nm and the highest was 2 nm. Extreme scaling transformations (greater than 5x) would exceed the limits of features commonly sampled in CEM500K, rendering invariance to such transformations useless. We expect that the superior robustness to variations in cellular EM data baked into CEM500K-moco should simplify the process of adjusting to new tasks. For example, when training a U-Net on a segmentation task, the parameters in the decoder will receive a consistent signal from the pre-trained encoder regardless of the orientation and other typical variations of the input image, presumably easing the learning burden on the decoder. For the same reason, we expect models to gain robustness to rare and random events such as experimental artifacts generated during sample preparation or image acquisition.

## CEM500K models learn biologically relevant features

Next, we assessed selectivity for objects of interest, that is, do these models learn something meaningful from cellular EM images? We created feature maps by appropriately upsampling the activations of each of the 2048 neurons in the last layer of the pre-trained ResNet50 and correlated these maps to the ground truth segmentations for three different organelles. In *Figure 3b*, activations of the 32 neurons most positively correlated with the presence of the corresponding organelle were averaged, scaled from 0 to 1 (displayed as a heatmap), and then binarized with a threshold of 0.3 (displayed as a binary mask). We observed that these derived heatmaps from the CEM500K-moco model shared a higher correlation with the presence of an organelle than features from the equivalent IN-moco model, irrespective of whether the organelle interrogated was ER, mitochondria, or nucleus. For the CEM500K-moco model, Point-Biserial correlation coefficients were 0.418, 0.680, and 0.888 for ER, mitochondria, and nucleus compared to 0.329, 0.608, and 0.803 for the IN-moco model. The segmentations created by binarizing the mean responses also have a greater IoU with ground truth segmentations (CEM500K-moco: 0.284, 0.517, and 0.887 for ER, mitochondria, and nucleus; IN-moco: 0.208, 0.325, and 0.790, respectively) for the model. Unexpectedly, features learned from ImageNet displayed some selectivity for mitochondria and nuclei, emphasizing the surprising transferability of features to domains that are seemingly unrelated to a model's training dataset. Nevertheless, it is clear that relevant pre-training, as is the case with CEM500K-moco, results in the model learning features that are meaningful in a cell biological context. The link between these results and the subsequent model's performance on downstream segmentation tasks is self-evident.

Pre-training on CEM500K encouraged the learning of representations that encode information about organelles. We analyzed how the model completed the MoCoV2 training task of matching differently transformed views of the same image. We first generated two different views of the same image by taking random crops and then randomly rescaling them. Then, we took one of the images in the pair and sequentially masked out small squares of data and measured the dot product similarity between the model's output on this occluded image and its output on the other image in the pair. Using this technique, called occlusion analysis, we were able to detect the areas in each image that were the most important for making a positive match (*Zeiler and Fergus, 2014*). Results are displayed as heatmaps overlaid on the occluded image (*Figure 3c*) and show, importantly, that without any guidance the model spontaneously learned to use organelles as 'landmarks' in the images, visible as 'hot spots' around such features. This behavior mirrors how a human annotator would likely approach the same problem: identify a prominent object in the first image and look for it in the second image. That these prominent objects should happen to be organelles is not coincidental as sample preparation protocols for electron microscopy are explicitly designed to accentuate organelles and membranes relative to other content. Thus, representations learned by CEM500K-moco pre-training display robustness to EM-specific image variations and selectivity for objects of interest, demonstrating that they should be well-suited to any downstream segmentation tasks.

With this understanding for how a model pre-trained with MoCoV2 on an EM-specific dataset might confer an advantage for EM segmentation tasks as compared to similar pre-training on a natural image dataset (ImageNet), we quantified this advantage by evaluating IoU improvements across the benchmark datasets. In addition to the CEM500K-moco and IN-moco pre-trained encoders, we also considered two alternative parameter initializations: ImageNet Supervised (IN-super)(*He et al., 2019*) and, as a baseline, random initialization. In contrast to results in *Figure 2c*, all encoder parameters for randomly initialized models were updated during training. Pre-trained models, as before, had their encoder parameters frozen to assess their transferability.

## Fully trained CEM500K models achieve state-of-the-art results on EM benchmarks

Results showing the measured percent difference in IoU scores against random initialization are shown in *Figure 4a*. For each benchmark, we applied the number of training iterations that gave the best performance for CEM500K-moco pre-trained models and averaged the results from five independent runs (see *Table 2*). Across the board, CEM500K-moco was the best initialization method with performance increases over random initialization ranging from 0.2% on the Lucchi++ benchmark to a massive 73% on UroCell; the mean improvement (excluding CREMI Synaptic Clefts) was 30%. For the short training schedule used, the baseline random initialization IoU score on the CREMI

**Table 2.** Comparison of segmentation IoU scores for different weight initialization methods versus the best results on each benchmark as reported in the publication presenting the segmentation task.

All IoU scores are the average of five independent runs. References listed after the benchmark names indicate the sources for Reported IoU scores.

| Benchmark | Training Iterations | Random Init. | IN-super | IN-moco | CEM500K-moco | Reported |
|---|---|---|---|---|---|---|
| All Mitochondria | 10000 | 0.587 | 0.653 | 0.653 | <u>0.770</u> | – |
| CREMI Synaptic Clefts | 5000 | 0.000 | 0.196 | 0.226 | <u>0.254</u> | – |
| Guay (*Guay et al., 2020*) | 1000 | 0.308 | 0.275 | 0.300 | <u>0.429</u> | 0.417 |
| Kasthuri++ (*Casser et al., 2018*) | 10000 | 0.905 | 0.908 | 0.911 | <u>0.915</u> | 0.845 |
| Lucchi++ (*Casser et al., 2018*) | 10000 | 0.894 | 0.865 | 0.892 | <u>0.895</u> | 0.888 |
| Perez (*Perez et al., 2014*) | 2500 | 0.672 | 0.886 | 0.883 | <u>0.901</u> | 0.821 |
| Lysosomes | – | 0.842 | 0.838 | 0.816 | <u>0.849</u> | 0.726 |
| Mitochondria | – | 0.130 | 0.860 | 0.866 | <u>0.884</u> | 0.780 |
| Nuclei | – | 0.984 | 0.987 | 0.986 | <u>0.988</u> | 0.942 |
| Nucleoli | – | 0.731 | 0.859 | 0.865 | <u>0.885</u> | 0.835 |
| UroCell | 2500 | 0.424 | 0.584 | 0.618 | <u>0.734</u> | – |

Synaptic Clefts benchmark was 0.000, making any % measurements of performance improvements meaningless. (The large foreground-to-background imbalance for synaptic clefts necessitates longer training schedules for the combination of randomly initialized models and unweighted binary cross-entropy loss). For ease of visualization, we assigned an IoU score of 0.2 for this dataset and calculated improvements based off of this score. Example 2D and 3D segmentations on the UroCell benchmark test set are shown in *Figure 4b*; we also display representative segmentations for selected labelmaps from all of the 2D-only benchmarks in *Figure 4c*. On the UroCell test set, all of the initialization methods except CEM500K-moco failed to accurately segment mitochondria in an anomalously bright and low-contrast region (example marked by a black arrow in *Figure 4b*). Indeed, CEM500K-moco also correctly identified features that the human annotator appears to have missed (example of missed mitochondrion, red arrow in *Figure 4c*). On average, IN-super and IN-moco achieved 11% and 14% higher IoU scores than random initialization, respectively. Parameters pre-trained with the unsupervised MoCoV2 algorithm thus appear to transfer better to new tasks than parameters pre-trained on the ImageNet supervised classification task (*He et al., 2019*). Crucially, the 14% average increase in IoU scores from CEM500K-moco over IN-moco reveals the advantage of pre-training on a domain-specific dataset. Thus, while it is clear that some of CEM500K-moco's improvement over random initialization is explained by pre-training with the MoCoV2 algorithm in general, most of the improvement comes from the characteristics of the pre-training data.

In addition to better IoU performance, pre-trained models converged more quickly. We found that models pre-trained with the MoCoV2 algorithm converged the fastest (*Figure 4d*, top). Within just 500 iterations, these models reach over 90% of their performance at 10,000 training iterations, and within only 100 iterations, they achieve over 80%. In some cases, 100 iterations required less than 45 s of training on our hardware, which included an Nvidia P100 GPU, making this approach more feasible for resource-limited work. We posit that the faster training associated with the MoCoV2 algorithm stems from the much lower magnitudes of feature activations, as observed in *Tian et al., 2019*, which facilitates training with higher learning rates. CEM500K-moco models trained marginally faster than IN-moco models. This speedup may have stemmed from CEM500K-moco's better robustness to the chosen data augmentations, reducing variance in the feature maps received by the trainable U-Net decoder. Overall, these results suggest a suitability of CEM500K-moco models for applications where rapid turnarounds for, say, a roughly accurate segmentation may be desired. In cases where more accurate segmentations are required, faster training as we see in *Figure 4d* reduces the amount of time needed for hyperparameter optimization.

Finally, the plot of average IoU scores over a range of training iterations showed that the performance of randomly initialized models leveled off after 5000 iterations, see *Figure 4d*, bottom. Previously, it has been observed that granted enough time to converge, randomly initialized models can

often achieve comparable results to pre-trained models (*He et al., 2018*), and we did observe this for the easiest benchmarks (Perez, Lucchi++, and Kasthuri++, data not shown). After 30,000 iterations of training on these benchmarks, the performance of randomly initialized models effectively reached parity with CEM500K-moco models. However, for the hard benchmarks, randomly initialized models never reached the average IoU scores measured at even just 500 training iterations for CEM500K-moco models. ImageNet pre-trained models, on the other hand, had the lowest average IoUs on easy benchmarks, but were better than random initialization for hard benchmarks. All of these observations align with expectations. Pre-trained models with frozen encoders only have $9 \times 10^6$ parameters to fit to the data. On easy benchmarks where overfitting is not a concern, this reduction in trainable parameters hurt ImageNet pre-trained models, but not CEM500K-moco models, since the latter were already pre-trained to EM data. On hard benchmarks, the regularization effects of having fewer trainable parameters are an advantage. Randomly initialized models continued to decrease training loss on hard benchmarks, yet those gains did not translate to increases in test set IoU, a signature of overfitting (data not shown). Overfitting may be avoided by smaller models with fewer trainable parameters, similar to the pre-trained models; however, this would require costly and slow additional engineering and hyperparameter optimization for each benchmark. Our results show that regardless of whether benchmarks are easy or hard, CEM500K-moco pre-trained models trained the fastest and achieved the best IoU scores. Indeed, these models outperformed the customized algorithms and training schemes presented as baselines for four of the benchmarks that we tested (by 3.0% on Guay, 8.6% on Kasthuri++, 1.2% on Lucchi++, and 10% on Perez), see *Table 2*. The All Mitochondria benchmark is a newly derived dataset and therefore has not been previously evaluated, but we show that it is a relatively challenging benchmark and suggest its use as a baseline for future comparisons. The remaining two benchmarks (CREMI Synaptic Clefts and UroCell) used special evaluation methods that were incompatible with our work (see 'Materials and methods'); instead, we present a representative visual comparison of our best results with those from the UroCell publication (*Appendix 1—figure 5*) showing a marked improvement in mitochondria (blue) and lysosome (red) 3D reconstructions. While ImageNet pre-trained models are broadly useful, our results show that for some EM segmentation tasks they perform worse than random initialization. For all the available benchmarks and the newly derived All Mitochondria benchmark, CEM500K-moco pre-training uniformly performed better than the current alternatives and we demonstrate here its reliability and effectiveness for EM-specific transfer learning.

## Discussion

CEM500K is a diverse, relevant, information-rich, and non-redundant dataset of unlabeled cellular EM images designed expressly to aid in the development of more robust and general DL models. Above all, two features distinguish CEM500K from other larger, publicly available EM datasets that make it superior for DL applications. First, it is derived from a far greater variety of tissue types, experimental conditions, and imaging techniques, resulting in models with less bias toward such specific variables. Second, it is condensed by aggressively deleting redundant and uninformative images; this improves model performance and renders CEM500K more accessible to users. By evaluating on seven benchmarks that represent different segmentation tasks and biological contexts, we demonstrate that, on average, models pre-trained on CEM500K performed better than those pre-trained on a dataset extracted from a single large EM volume (Bloss). Remarkably, the targeted removal of 90% of the images from the original corpus of data to generate CEM500K returned a significant increase in the quality of pre-trained parameters as measured by segmentation IoU scores.

This raises the question of what the nature and extent of dataset curation should be: If a target segmentation task contains data from a particular biological context, should the pre-training dataset be curated specifically for that context? And would pre-training on the task data alone result in adequate models? Our results suggest that the benefits from curating the pre-training dataset for a particular context are minimal. Pre-training exclusively on images of mouse brain tissue (Bloss) did not improve performance over CEM500K on benchmarks from that same tissue (see *Figure 2e*). Pre-training exclusively on images from a target dataset (say, for a segmentation task) is possible but would have limited utility as it nullifies the advantages introduced by dataset heterogeneity. Further, we speculate that as dataset size decreases, it becomes more likely that a model will overfit to the pre-training task and learn image features that are irrelevant for other downstream tasks (*Tian and*

*Sun, 2020*; *Minderer et al., 2020*). Other unsupervised pre-training algorithms that work for smaller datasets and/or larger benchmark datasets would be needed to determine the appropriate curation approach.

Regardless, we have shown here that parameters trained on CEM500K are a strong and general-purpose starting point for improving downstream segmentation models. U-Nets pre-trained on CEM500K significantly outperformed randomly initialized U-Nets on all of the segmentation benchmarks that we tested, with the largest improvements corresponding to the most difficult benchmarks. Impressively, such pre-trained models achieved state-of-the-art IoU scores on all benchmarks for which comparison with previous results was possible. The only variables tuned were the number of training iterations and data augmentations. Use of CEM500K pre-trained models by the EM community may reveal that further tuning of hyperparameters or unfreezing of the U-Net's encoder parameters could further boost performance.

Our work focused on the application of CEM500K for transfer learning. This decision was informed by the current status of DL research for cellular EM, where, typically, segmentation tasks are performed by models trained on a few labeled examples (*Funke et al., 2019*; *Guay et al., 2020*; *Perez et al., 2014*; *Žerovnik Mekuč et al., 2020*; *Casser et al., 2018*; *CREMI, 2016*). In general, pre-trained parameters have been shown to guide downstream models to converge to more general optima than they would from random initialization (*Huh et al., 2016*; *Yosinski et al., 2014*; *Neyshabur et al., 2020*). As the number of examples in the training dataset increases the generalization benefits from transfer learning start to diminish (gains in training speed are retained) (*Zoph, 2020*; *He et al., 2018*). Therefore, while unsupervised pre-training on CEM500K for transfer learning has demonstrably high utility for the common paradigm of 'train on labeled ROIs/infer labels for the whole dataset', currently it cannot solve the problem of creating general segmentation models that reliably segment features of interest for data generated by novel experiments. However, using CEM500K as seed data provides a path forward for tackling this much more difficult challenge. With $0.5 \times 10^6$ uniquely informative images representing approximately six hundred 3D and ten thousand 2D images corresponding to more than 100 completely unrelated biological projects, CEM500K is to our knowledge the most comprehensive and diversified resource of cellular EM images. Annotating images from CEM500K (or identifying them as negative examples) will enable the creation of new task-specific training datasets with substantially more variety than previously available. Models trained on such datasets will likely be better equipped to handle data from new microscopes, biological contexts, and sample preparation protocols. Moreover, each image chosen for annotation from CEM500K is likely to be uniquely informative for a model because of the extensive deduplication and filtering pipeline that we have created and used here, and which we share for future work by the community.

The available benchmark datasets that we chose are a reflection of common applications of DL to cellular EM data, but they do not cover the full scope of possible segmentation tasks. In particular, all but one of the benchmarks involved the annotation of mitochondria and three of the seven were from mouse brain tissue. We observed that benchmark variety is essential to identify biases in pre-trained parameters and that difficult tasks are a necessary and stringent test of pre-training algorithms or datasets. For example, visual inspection of the label maps in *Figure 4c* makes it obvious that our results leave little room for improvement on relatively easy (and 2D only) benchmarks like Lucchi++, Kasthuri++, and Perez, suggesting that going forward, new and more challenging benchmarks will be required.

Additionally, we only tested semantic and not instance segmentation (i.e. all objects from one class share one label). We made this decision in order to avoid the more complex model architectures, postprocessing, and hyperparameters that usually accompany instance segmentation (*Heinrich et al., 2018*; *Funke et al., 2019*; *Januszewski et al., 2018*). Focusing on simple end-to-end semantic segmentation tasks emphasizes the effects of pre-training and eliminates the possibility that non-DL algorithms could confound the interpretation of our results. 2D models work well for semantic segmentation in both 2D and 3D (our 2D models beat the state-of-the-art results set by 3D models on some of the benchmarks, see *Table 2*) and from a computational standpoint, running training and inference on 2D data typically requires fewer resources. Therefore, we believe that 2D pre-trained parameters are broadly useful for cellular EM researchers, especially for laboratories with limited access to high-performance computing. For some complex tasks like 3D instance segmentation, an important and common task in cellular EM connectomics research, 2D models may perform

poorly. Unsupervised pre-training on 3D data is currently an underexplored research area and algo-rithms like MoCoV2 should work for pre-training 3D models. The creation of a 3D image dataset similar to the resource reported here is likely to be of great use to the nascent volume EM community.

The goal of this work is to begin the process of creating a data ecosystem for cellular EM images and datasets. CEM500K will be a valuable resource for experimenting with and taking advantage of the latest developments in DL research, where access to troves of image data is usually taken for granted. To further increase its utility, more data from uncommon organisms, tissue and cell types, sample preparation protocols, and acquisition parameters will be needed. In the current state, the dataset is still heavily skewed to a few common organisms like mice and tissues like brain, and it is clear that there is much room for greater sampling and heterogeneity (*Appendix 1—figure 6*). We hope that other researchers will consider using the curation tools that we developed in this work to contribute to CEM500K. The dataset has been made available for download on EMPIAR (ID 10592). The massive reduction in dataset size from curation makes the sharing of data relatively quick and easy; moreover, the elimination of 3D context from volume EM datasets ensures that the shared data can only reasonably be used for DL applications. Similar to pre-training on natural images, we expect that the quality of the pre-trained parameters for transfer learning will improve logarithmi-cally as CEM500K grows (*Mahajan, 2018*). In the meantime, the pre-trained parameters that we release here can serve as the foundation for rapidly prototyping and building more general segmen-tation models for cellular EM data.

# Materials and methods

## Dataset standardization

Datasets generated from microscopes in our lab were already in the desired standardized format: 8-bit unsigned volumes or 2D tiff images. Publicly available EM data are in a variety of file formats and data types; these datasets were individually reformatted as needed to match the formatting of our internal datasets. Importantly, data from each of the seven benchmarks we tested were included as well but comprised less than 0.1% of the dataset. To reduce the memory requirements of large 3D volumes, datasets were downsampled such that no individual dataset was larger than 5 GB (affecting only seven of the total 591 image volumes). The majority of 3D datasets included metadata of their image resolutions; isotropic and anisotropic volumes were thus automatically identified and proc-essed differently. For all isotropic voxel data and for any anisotropic voxel data in which the z resolu-tion was less than 20% different from the x and y resolutions, 2D cross-sections from the xy, xz, and yz planes were sequentially extracted. Anisotropic voxel data with a greater than 20% difference in axial versus lateral resolutions were only sliced into cross-sections in the xy plane. At this point, all of the gathered image data was in the format of 2D tiff images, though with variable heights and widths. These images were cropped into 224 × 224 patches without any overlap. If the image's width or height was not a multiple of 224, then crops from the remaining area were discarded if either of their dimensions were less than 112.

Separately, additional 2D images available through the Open Connectome Project were col-lected. As these volumes were too large to reasonably download and store (tens of TB), Cloud Vol-ume API was used to randomly sample 1000 2D patches from the xy planes of each available dataset. These extracted patches were already of the correct size and format, therefore no further processing was required. This corpus of $5.3 \times 10^6$ 2D patches constitutes 'CEMraw'. Certain data-sets were not accessible with this method and were therefore not included in the final version of CEMraw (see Supplementary Materials). The 'Bloss baseline' dataset (*Bloss et al., 2018*) was also extracted and generated with this method; however, $1 \times 10^6$ patches were collected from that sin-gle data volume to roughly match the number of images in CEMraw (*Appendix 1—figure 4*).

## Deduplication

To remove duplicate patches, image hashes for all $5.3 \times 10^6$ images in CEMraw were calculated. Dif-ference hashes gave the best results of all the hashing algorithms tested (*Kind of Like That, 2013*). A hash size of 8 results in a 64-bit array to encode each 224 × 224 image. The similarity of two images was then measured by the Hamming distance between their hashes. A pairwise comparison

of all $5.3 \times 10^6$ hashes was not computationally feasible or meaningful. Instead, hashes belonging to the same 2D or 3D source dataset were compared. For a 64-bit hash, distances range from 0 to 64. Sets of hashes with a distance <12 (distance cutoff chosen by visual inspection of groups) between them were considered a group of near-duplicates. All but one randomly chosen image from each group were dropped (*Figure 1b*). Together, the resulting $1.1 \times 10^6$ images constitute a deduplicated dataset or 'CEMdedup'.

## Uninformative patch filtering

A random subset of 14,000 images from CEMdedup were manually labeled either informative or uninformative. The criteria for this classification process were informed by the hyperparameters of the MoCoV2 pre-training algorithm, which takes random crops as small as 20% of an area of an image. For an image that is only 20% informative, there is a 30% chance that such a randomly drawn crop will be completely uninformative, and this fraction increases exponentially for images less than 20% informative (*Appendix 1—figure 7*). Therefore, 20% was chosen as the cutoff for manual labeling. Concretely, this means that images with 80% or more of their area occupied by uniform intensity structures like nuclei, cytoplasm, or resin are classified as uninformative. Other criteria included whether the image was low-contrast, displayed reconstruction artifacts (not sample preparation or image acquisition artifacts), or contained non-cellular objects as determined by a human annotator. A breakdown of the frequency of traits present in a subset of uninformative patches is shown in *Appendix 1—figure 1a*.

2000 labeled images were set aside as a test set and the remaining 12,000 were used as training data for a model classifier: a ResNet34 pre-trained on ImageNet. The fourth layer of residual blocks and the classification head of the model were fine-tuned for 30 epochs on a P100 GPU with the Adam optimizer and a learning rate of 0.001. A Random Forest classifier trained on four image-level statistics (the standard deviations of the local binary pattern [*Ojala et al., 2002*] and image entropy, the median of the geometric mean, and the mean value of a canny edge detector [*Canny, 1986*]) was also tested. These features were chosen from a larger superset based on their measured importance. The performance for the two classifiers is shown in *Appendix 1—figure 1b*. The DL model was used to create CEM500K with a confidence threshold set at 0.5. Since the benchmark data themselves were put through the curation pipeline, approximately 5% of images in CEM500K are from one of the six benchmark datasets; we show that the presence of this small fraction has a marginal effect on performance (*Supplementary file 2*).

## Momentum contrast pre-training

For unsupervised pre-training, the Momentum Contrast (MoCoV2) algorithm (*Raghu et al., 2019*; *Tian et al., 2019*) was used. A schematic of a single step in the algorithm is shown in *Appendix 1—figure 3a*. Pre-training was completed on a machine with four Nvidia V100 GPUs using a batch size of 128 and queue length of 65,536. The initial learning rate was set to 0.015 and divided by 10 at epochs 120 and 160. In addition, 360° rotations and Gaussian noise with a standard deviation range of $1 \times 10^{-5}$ to $1 \times 10^{-4}$ were added to the data augmentations. All other hyperparameters and data augmentations were left as the defaults presented in *Tian et al., 2019*. For pre-training comparisons between different EM datasets, that is the three subsets of CEM plus Bloss (*Figure 2d,e*), $4.5 \times 10^5$ total parameter updates (iterations) were run for each model, which is equivalent to 120 passes (epochs) through all the images in CEM500K. The average training time for each of these models was 2.5 days. The final pre-trained parameters generated for results shown in *Figure 4b,c* were trained on CEM500K for an additional 80 epochs: a total of 200 epochs and 4 days of training.

## U-Net segmentation architecture

Our implementation was similar to the original implementation of the U-Net, except that the encoder was replaced with a ResNet50 model (*Appendix 1—figure 3b*). When using pre-trained models in these experiments, all parameters in the encoder were frozen such that no updates were made during training. Randomly initialized encoders were tested with both frozen and unfrozen parameters. The random number generator seed was fixed at 42 such that any randomly initialized parameters in either the U-Net encoder or decoder would be the same in every experiment.

## Benchmark segmentation tasks

The One Cycle Policy and AdamW optimizer with maximum learning rate 0.003, weight decay 0.1, batch size 16, and (binary) cross-entropy loss were used for all benchmarks (*Loshchilov and Hutter, 2017*; *Smith, 2018*). For the Guay and Urocell benchmarks, which required multiclass segmentation, the cross-entropy loss was weighted by the prevalence of each class; we observed that this yielded better IoU scores on the Guay validation dataset (*Supplementary file 3*). Classes that accounted for less than 10% of all pixels in the dataset were given a weight of 3, those that accounted for more than 10% were given a weight of 1, and all background classes were given a weight of 0.1. Data augmentations included randomly resized crops with scaling from 0.08 to 1 and aspect ratio from 0.5 to 1.5, 360° rotations, random 30% brightness and contrast adjustments, and horizontal and vertical flips. For the Guay benchmark, and consequently the All Mitochondria benchmark, Gaussian noise with a variance limit of 400–1200 and Gaussian blur with a maximum standard deviation of 7 were also added. The decision to add more data augmentations for these benchmarks was made in response to observed overfitting on the Guay benchmark validation dataset (*Supplementary file 3*). Lastly, different crop sizes were used for each benchmark: 512 × 512 for Guay, CREMI, Synaptic Cleft, Kasthuri++, and Lucchi++; 480 × 480 for Perez; and 224 × 224 for UroCell and All Mitochondria.

To create 3D segmentations for the UroCell, Guay, and CREMI Synaptic Cleft, test sets we used either orthoplane or 2D stack inference following *Conrad et al., 2020*. Briefly, in 2D stack inference the model only makes predictions on xy cross-sections; in orthoplane inference, the model makes predictions on xy, yz, and xz cross-sections and the confidence scores are averaged together. Orthoplane inference was used for the UroCell test set because its test volume has isotropic voxels. Because both the Guay and CREMI Synaptic Cleft test volumes are anisotropic, we used 2D stack inference instead.

Evaluation generally followed the details given in the publication that accompanied the benchmark. First, test images in the Perez datasets did not have labels for all instances of an object; for example, only one nucleus was labeled in an image containing two nuclei. To circumvent this problem, we ignored areas in the predicted segmentations that did not coincide with a labeled instance in the ground truth. Second, the UroCell benchmark was evaluated in previous work by averaging K-Fold cross-validation results on five unique splits of the five training volumes such that each training volume was used as the test set once. The authors also excluded pixels on the boundary of object instances both when training and when calculating the prediction's IoU with ground truth. Here, a simpler evaluation was run on a single split of the data with four volumes used for training and one volume used for testing. To eliminate small regions of missing data, we cropped two of the five volumes along the y axis (fib1-0-0-0.nii.gz, the test volume, by 12 pixels and fib1-1-0-3.nii.gz by 54 pixels). Third, the test data for the CREMI Synaptic Cleft benchmark is not publicly available and the challenge uses a different evaluation metric than IoU. Therefore, volumes A and B were used exclusively for training and IoU scores were evaluated on volume C.

## Mean firing rate

Following *Goodfellow et al., 2009*, neuron firing thresholds were determined by passing 1000 images of randomly sampled noise through each pre-trained ResNet50 model and calculating the 99th percentile of responses. In our experiments, only the neurons in the output of the global average pooling layer were considered such that there were 2048. Responses to 100 randomly selected images from CEM500K were then recorded over a range of distortion strengths. For each neuron, the set of undistorted images that activated the neuron near maximally (over the 90th percentile), called Z, was determined. A set containing versions of all images in Z with a particular distortion applied is called Z'. Any neuron that responded to images in Z less strongly than the neuron's firing threshold were ignored as they are not selective for features observed in the test images. However, for all remaining neurons, the firing rate at a particular distortion strength is calculated as the number of images in Z' that activate the neuron over its firing threshold divided by the number of images in Z. The mean firing rate to a particular distortion is then the average of firing rates for any of the 2048 neurons that were selective enough to be considered.

## Feature selectivity

To measure feature selectivity, we first manually segmented three organelles (ER, mitochondria, nucleus) in three images. By construction, the ResNet50 architecture downsamples an input image by 32. For thin and small organelles like ER, the final feature maps were too coarse to accurately show the localization of responses. Therefore, we eliminated the last four downsampling operations such that the output feature map was only 2x smaller than the input. Following similar logic, we eliminated the last two downsampling operations for mitochondria and the last downsampling operation for nuclei – 8x and 16x smaller than the input images, respectively. For all organelles, these differently downsampled feature maps were resized to match the dimensions of the input image ($224 \times 224$) and then each feature map was compared against the ground truth labelmap by Point Biserial correlation. A simple average of the 32 most correlated feature maps was then overlaid on the original image as the mean response. Drawing a threshold at 0.3 yielded the binary segmentations.

## Occlusion analysis

Typically, occlusion analysis measures the importance of regions in an image to a classification task (*Zeiler and Fergus, 2014*). In our experiments, importance was measured as a function of the dot product similarity between the feature vectors output by the global average pooling layer of a ResNet50 for an image and its occluded copy. Sequential regions of $61 \times 61$ pixels spaced every 30 pixels (in both x and y dimensions) were zeroed out in each image. Region importance to the similarity measurement was then normalized to fall in the range 0–1 and overlaid on the original image.

## Acknowledgements

We thank the creators of the benchmarks and other datasets for making the image data freely available to the community. We thank Patrick Friday for help with running some of the models described herein, members of the CMM for contributing EM images, and FNL and NCI colleagues for critical comments on this manuscript. This work utilized the computational resources of the NIH HPC Biowulf cluster (http://hpc.nih.gov). This project has been funded in whole or in part with Federal funds from the National Cancer Institute, National Institutes of Health, under contract no. 75N91019D00024. The content of this publication does not necessarily reflect the views or policies of the Department of Health and Human Services, nor does mention of trade names, commercial products, or organizations imply endorsement by the U.S. Government.

## Additional information

### Funding

| Funder | Grant reference number | Author |
| --- | --- | --- |
| National Cancer Institute | Contract No. 75N91019D00024 | Ryan Conrad Kedar Narayan |

The funders had no role in study design, data collection and interpretation, or the decision to submit the work for publication.

### Author contributions

Ryan Conrad, Conceptualization, Data curation, Software, Formal analysis, Validation, Visualization, Methodology, Writing - original draft, Writing - review and editing; Kedar Narayan, Conceptualization, Resources, Data curation, Supervision, Funding acquisition, Visualization, Writing - review and editing

### Author ORCIDs

Kedar Narayan (iD) https://orcid.org/0000-0001-7982-6494

### Decision letter and Author response
Decision letter https://doi.org/10.7554/eLife.65894.sa1
Author response https://doi.org/10.7554/eLife.65894.sa2

## Additional files

### Supplementary files

- Source data 1. Details of image datasets acquired from external sources.
- Source data 2. Zipped folder containing .xl files for *Figure 1*, *2* and *4* source data.
- Supplementary file 1. Details of benchmarks used in this paper.
- Supplementary file 2. IoU scores for pre-training with CEM500K after removing benchmark data from pre-training dataset .
- Supplementary file 3. IoU scores on Guay benchmark using different hyperparameter choices.
- Transparent reporting form

### Data availability

The CEM500K dataset, metadata, and pre-trained weights available at: https://www.ebi.ac.uk/pdbe/emdb/empiar/entry/10592/. The full code for curation pipeline is available at https://git.io/JLLTz. Links to publicly available benchmark datasets are provided in Supplementary File 1.

The following dataset was generated:

| Author(s) | Year | Dataset title | Dataset URL | Database and Identifier |
|---|---|---|---|---|
| Conrad R, Narayan K | 2020 | CEM500K | https://www.ebi.ac.uk/pdbe/emdb/empiar/entry/10592/ | Electron Microscopy Public Image Archive, EMPIAR-10592 |

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

## Appendix 1

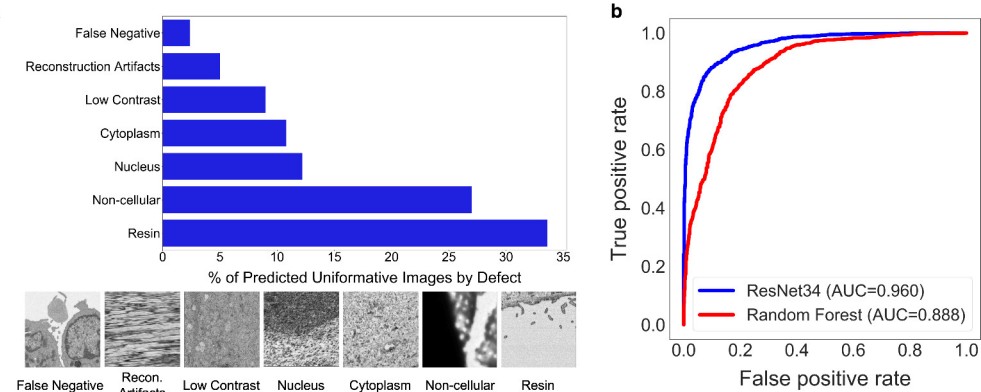

**Appendix 1—figure 1.** Deduplication and image filtering. (**a**) Breakdown of fractions (top) and representative examples (bottom) of patches labeled 'uninformative' by a trained deep learning (DL) model based on defect (as determined by a human annotator). (**b**) Receiver operating characteristic curve for the DL model classifier and a Random Forest classifier evaluated on a holdout test set of 2000 manually labeled patches (1000 informative and 1000 uninformative).

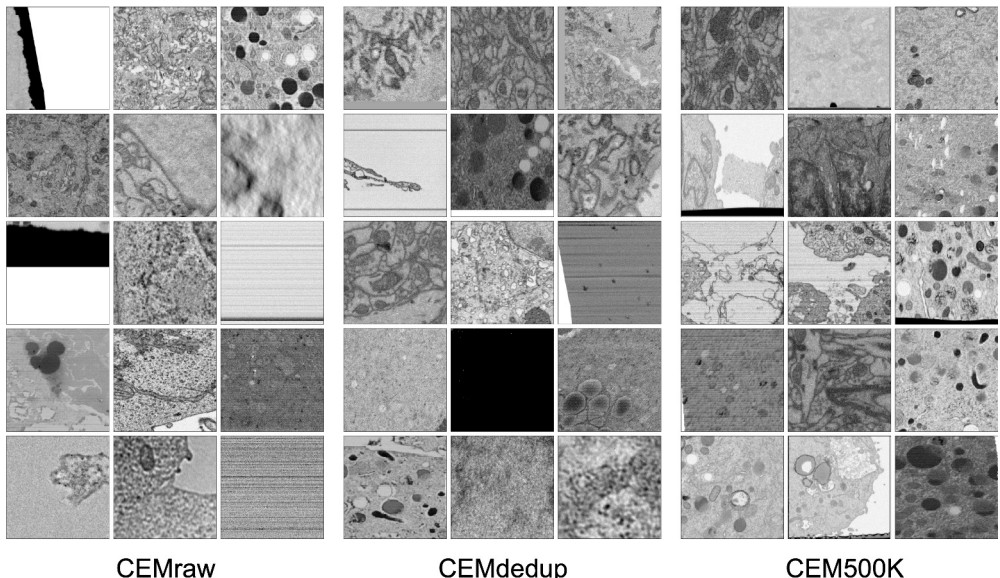

**Appendix 1—figure 2.** Randomly selected images from CEMraw, CEMdedup, and CEM500K.

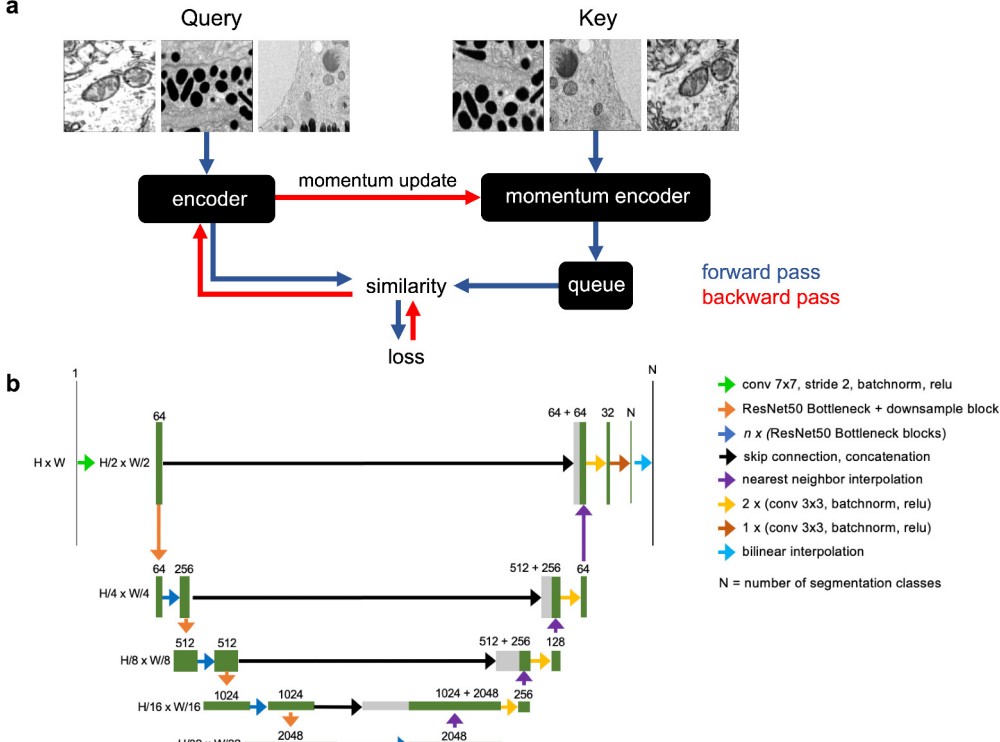

**Appendix 1—figure 3.** Schematics of the MoCoV2 algorithm and UNet-ResNet50 model architecture. (**a**) Shows a single step in the MoCoV2 algorithm. A batch of images is copied; images in each copy of the batch are independently and randomly transformed and then shuffled into a random order (the first batch is called the *query* and the second is called the *key*). *Query* and *key* are encoded by two different models, the *encoder* and *momentum encoder,* respectively. The encoded *key* is appended to the *queue.* Dot products of every image in the *query* with every image in the *queue* measure similarity. The similarity between an image in the *query* and its match from the *key* is the signal that informs parameter updates. More details in *He et al., 2019*. (**b**) Detailed schematic of the UNet-ResNet50 architecture.

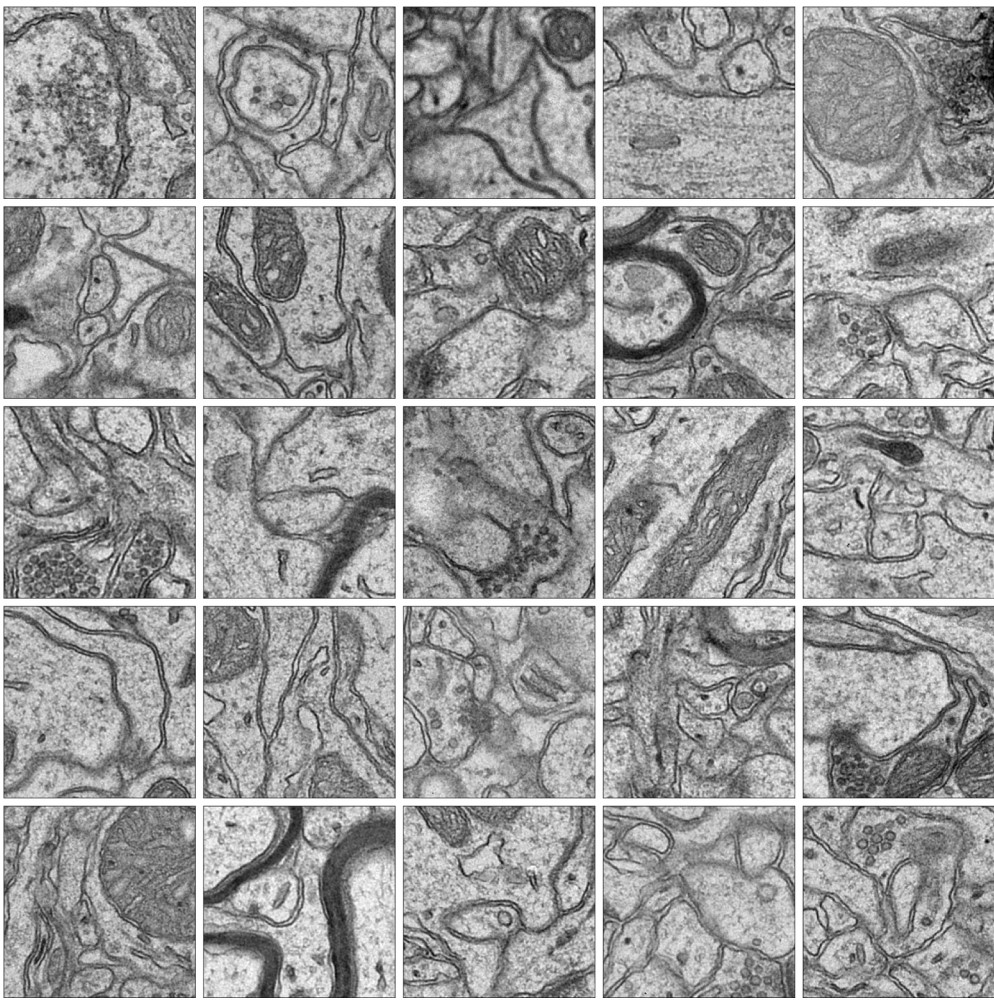

**Appendix 1—figure 4.** Randomly selected images from the *Bloss et al., 2018* pre-training dataset.

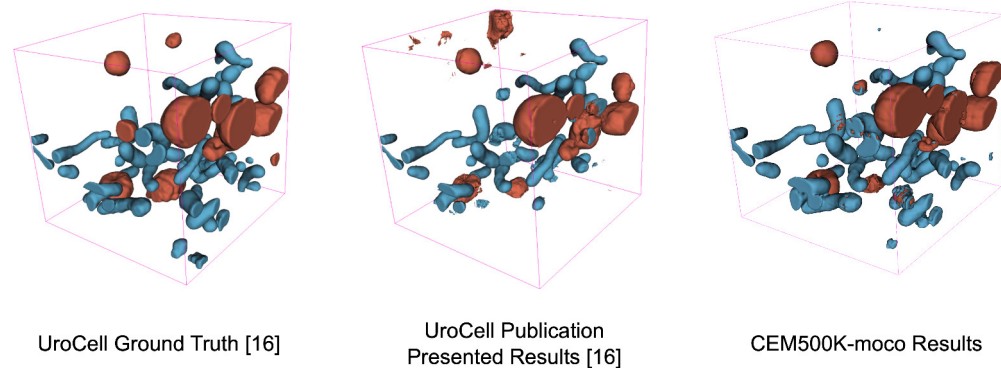

UroCell Ground Truth [16]

UroCell Publication
Presented Results [16]

CEM500K-moco Results

**Appendix 1—figure 5.** Visual comparison of results on the UroCell benchmark. The ground truth
and Authors' Best Results are taken from the original UroCell publication (*Žerovnik Mekuč et al.,
2020*). The results from the CEM500K-moco pre-trained model have been colorized to
approximately match the originals; 2D label maps were not included in the UroCell paper.

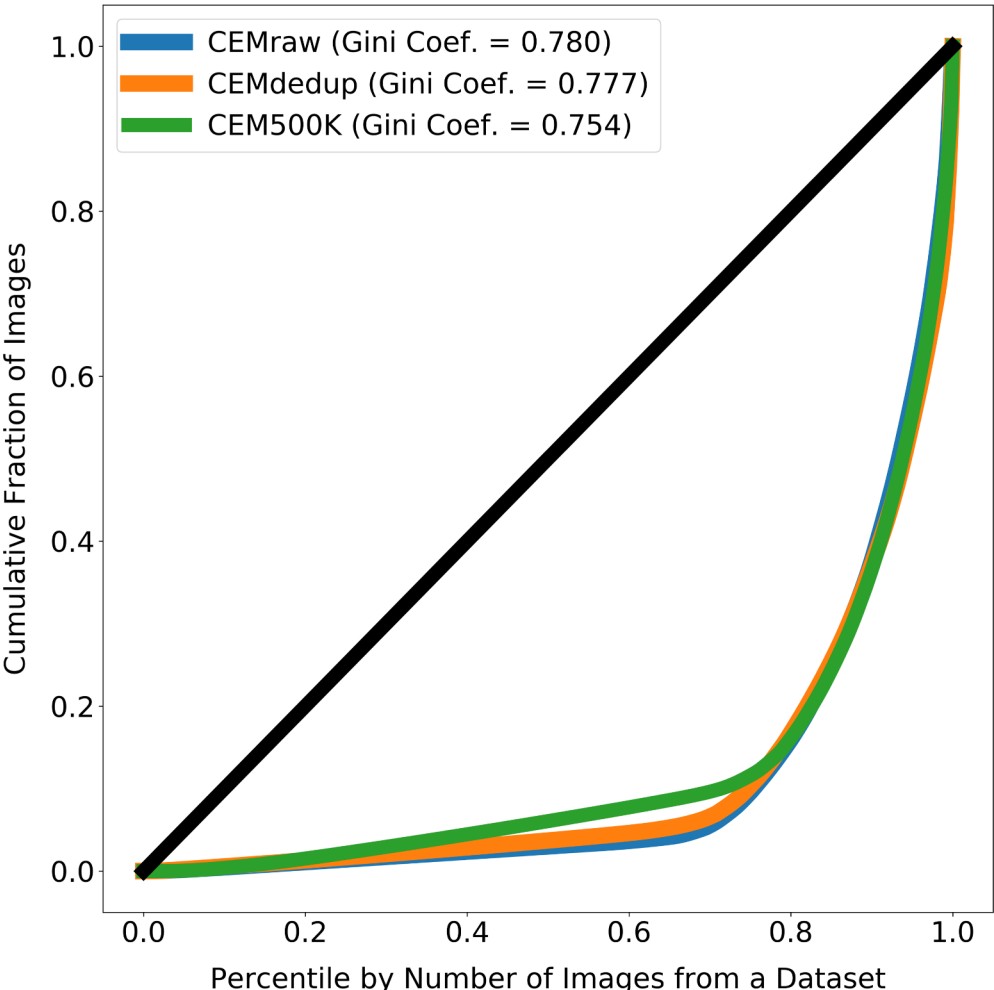

**Appendix 1—figure 6.** Images from source electron microscopy (EM) volumes are unequally represented in the subsets of CEM. The line at 45° shows the expected curve for perfect equality between all source volumes (i.e. each volume would contribute the same number of images to CEMraw, CEMdedup, or CEM500K). Gini coefficients measure the area between the Lorenz Curves and the line of perfect equality, with 0 meaning perfect equality and 1 meaning perfect inequality. For each subset of cellular electron microscopy (CEM), approximately 20% of the source 3D volumes account for 80% of all the 2D patches.

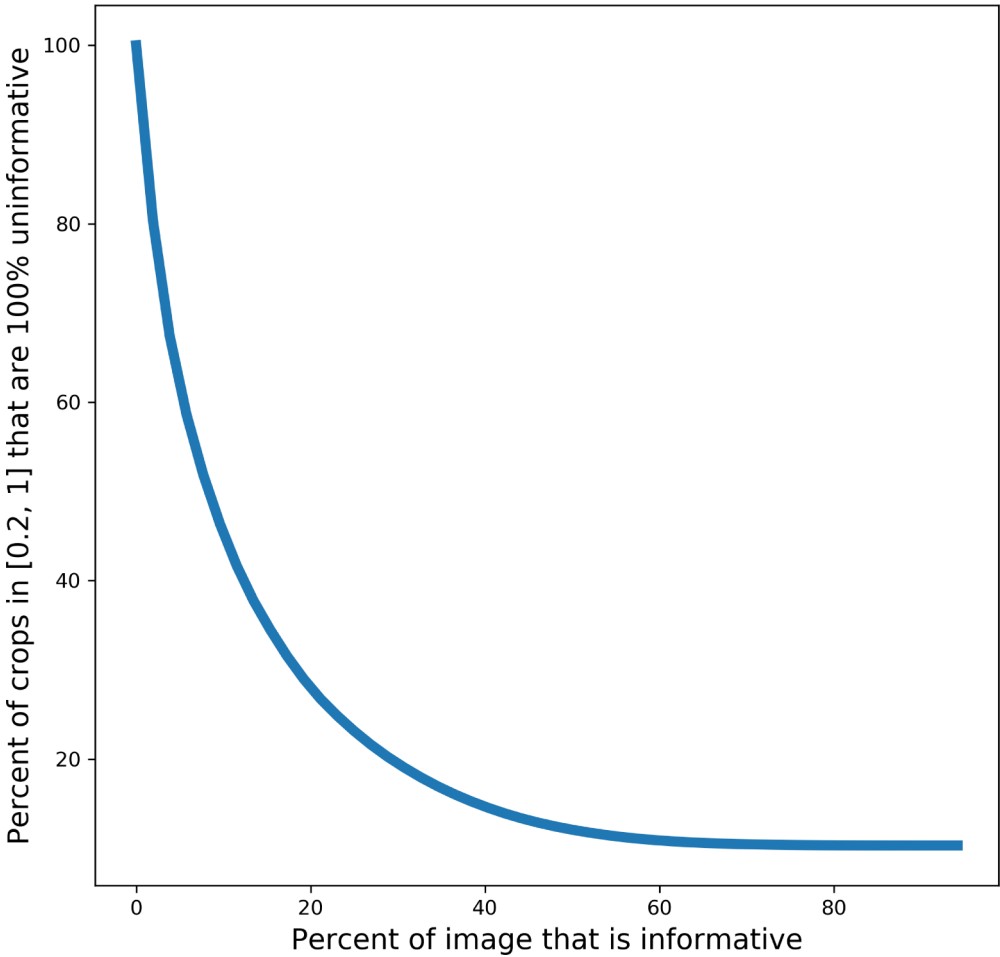

**Appendix 1—figure 7.** Plot showing the percent of random crops from an image that will be 100% uninformative based on the percent of the image that is informative.

