## [Decision Letter]

**Acceptance summary:**

This manuscript describes the curation of a training dataset that will be an important resource for developers of new segmentation and deep-learning algorithms for electron microscopy data. The small size of the dataset makes it easy to use, and its broad range of image modalities ensures that the model will be applicable in many situations, making it very useful for the community.

**Decision letter after peer review:**

Thank you for submitting your article "CEM500K – A large-scale heterogeneous unlabeled cellular electron microscopy image dataset for deep learning" for consideration by *eLife*. Your article has been reviewed by three peer reviewers, including Nikolaus Grigorieff as the Reviewing Editor and Reviewer #1, and the evaluation has been overseen by Anna Akhmanova as the Senior Editor. The following individual involved in review of your submission has agreed to reveal their identity: Stephan Saalfeld (Reviewer #2).

Essential Revisions:

1) This is an excellent manuscript. The authors establish a broadly usable dataset, and designed and conducted a strict and sound evaluation study. The paper is well written, easy to follow and overall well balanced in how it discusses technical details and the wider impact of this study. However, the work focuses on pixel classification on 2D images. This is not clear in the beginning and only discussed fairly late in the manuscript where 3D approaches are generally dismissed for no other reason than that they are not applicable to 2D images and more expensive. This dismissal is invalid and unnecessary. The authors should mention their focus on 2D images earlier in the manuscript, explain why this approach is taken, and what would have to be changed to include 3D data.

Reviewer #1 (Recommendations for the authors):

1) The authors should clarify that none of the benchmark data have been included in the CEM500K training set.

2) The authors should include a supplementary table for the cross-validation scores they obtained to validate their trained CEM500K-moco model.

3) Given the relatively small size of the CEM500K dataset, how much of a concern is overfitting?

4) Can the authors comment on the resolution of features that are likely relevant in their CEM500K-moco model? What are the prospects of using MoCoV2 to recognize higher-resolution features in cryo-EM images?

Reviewer #2 (Recommendations for the authors):

1) This work clearly targets pixel classification on 2D images, however the 2D focus is not clear in the beginning and only discussed late (Discussion) where 3D approaches are generally dismissed for no other reason than that they are not applicable to 2D images and more expensive. This dismissal is invalid and also unnecessary and should be removed. The 2D focus should become obvious in the Abstract and Introduction of the paper since the term "image" is used for nD data in the community.

2) Introduction, third paragraph: add paragraph.

3) The hashing algorithm used for identifying near duplicates (https://www.hackerfactor.com/blog/index.php?/archives/529-Kind-of-Like-That.html) may not be optimal but this probably doesn't do much harm as its main purpose is to remove nearby sections from 3D volumes.

4) Could CEMdedup be useful for tasks that contain empty resin or nuclei?

5) Conclusion: Pretraining dataset should include data relevant for the benchmark task. Does unrelated data necessarily make the benchmark performance better? How would CEM500K + Bloss pre-training perform for the mouse data?

6) CEM500K MoCoV2 training included 360° rotations, i.e. the increased rotation invariance compared to ImageNet is by design of the training, not necessarily the data.

7) Subsection “CEM500K models are largely impervious to meaningful image augmentations”: Why would models gain robustness to rare and random events such as artifacts? Weren't such images excluded from the dataset?

8) Subsection “CEM500K models learn biologically relevant features”: Could this also be just high contrast/high gradient regions which makes sense for the given training task? I.e. is the focus on objects of interest may be a trivial consequence of objects of interest being the only thing present/distinguishable in the image?

9) Subsection “Fully trained CEM500K models achieve state-of-the-art results on EM benchmarks”: The random initialization score for CREMI of 0.0 is surprising. Is this an indication that the training setup or this particular network architecture have other issues that are unrelated to the pre-training? Others have successfully trained synapse detection networks without pre-training.

10) Discussion, second paragraph: This is not correct for CREMI which is data from the large FAFB dataset, and also itself contains more data than necessary in the padded volumes available for download. I do not find it necessary to perform this experiment but it is ok to simply say that this hasn't been done without falsely asserting that it wasn't possible.

11) I do not understand this statement: "[…] moreover, the elimination of 3D context from volume EM datasets ensures that the shared data can only reasonably be used for DL." Is it necessary?

12) Materials and methods: The CREMI synaptic cleft evaluation metric is public including code to execute it. Ground truth for test data, however, is secret. Did you submit your results for evaluation? Like above, it is not necessary to do this but the statement "[…] the training and test datasets did not have an official evaluation metric, and the ground truth segmentations were not publicly available[…]" is not correct. May be replace with "The test data for the CREMI Synaptic Cleft benchmark is not publicly available and the challenge uses a different evaluation metric than IoU. Therefore[…]".

Thank you very much for this wonderful work. It's been a pleasure to read this manuscript.

Reviewer #3 (Recommendations for the authors):

– Would you please clarify the statement “a model's ability to generalize to unseen images”? I would like to see more clarification regarding this statement as it seems that it ignores the definition of transfer learning problems. Transfer learning techniques are applied where there exists lack of data. Therefore, providing datasets with large numbers of data does not seem to be a good solution for these problems though it is still a valuable dataset for many problems.

– In the first part of the paper, it is referred to "six publicly available dataset". I know that you name them in the later part of the paper but it is a question for the reader. It would be better to name them in these sections as well.

– It would be better if the minimum amount of outperformance is defined when it is presented "significantly outperformed randomly initialized".

– Is there any reference/proof for the following claim: "Although it is currently impossible to determine a priori what data will be useful for a model, we expect that this removal of significant redundancies in the image dataset is unlikely to result in the loss of meaningful information for DL model training."

– What would happen if you keep the duplicate images and use MoCoV2 algorithm?

– How about not applying the MoCoV2 algorithm and using other algorithms? Have you considered ablation studies in this regard?

– The following statements are unclear and I appreciate it if you clarify them more.

1) "These results emphasize the necessity of evaluating deep learning algorithms and pre-training datasets on multiple benchmarks before drawing conclusions about their quality."

2) “We expect that the superior robustness to variations in cellular EM data baked into CEM500K-moco should simplify the process of adjusting to new tasks.”

---

## [Author Response]

Essential Revisions:1) This is an excellent manuscript. The authors establish a broadly usable dataset, and designed and conducted a strict and sound evaluation study. The paper is well written, easy to follow and overall well balanced in how it discusses technical details and the wider impact of this study. However, the work focuses on pixel classification on 2D images. This is not clear in the beginning and only discussed fairly late in the manuscript where 3D approaches are generally dismissed for no other reason than that they are not applicable to 2D images and more expensive. This dismissal is invalid and unnecessary. The authors should mention their focus on 2D images earlier in the manuscript, explain why this approach is taken, and what would have to be changed to include 3D data.

Thank you for your positive comments, we now explicitly define CEM500K as a 2D image dataset in the Abstract and Introduction. We have also edited and clarified the “2D vs. 3D” argument in the Discussion.

Reviewer #1 (Recommendations for the authors):1) The authors should clarify that none of the benchmark data have been included in the CEM500K training set.

– The benchmark data, which represented ~5% of the total data was indeed included in the CEM500K training set for the results reported in Figures 2, 4. We did this to avoid the confusion of reporting results pre-trained without the benchmark data and releasing weights pre-trained with the benchmark data. The manuscript has been edited to make this point clear.

– However, the reviewer does bring up a fair point. To test whether including benchmark data misrepresents the quality of the pre-trained weights for unseen data, we had previously run experiments in which benchmark data was excluded; we saw both increases and decreases in IoU scores (average 0.6% decrease) depending on the dataset. Importantly, this comparison is not perfect: CEM500K without the benchmark data had ~24,000 fewer images, the same 200 epochs of training therefore included 37,000 fewer parameter updates (iterations).

– We have included a new table of these results, Supplementary file 2.

2) The authors should include a supplementary table for the cross-validation scores they obtained to validate their trained CEM500K-moco model.

– Thank you for noting this omission. We used cross-validation for the Guay benchmark dataset; none of the other benchmark datasets included predefined validation data splits and the accompanying publications did not report cross-validation results. Note this lack of data splits also informed our choice to use a fixed set of hyperparameters for all other benchmarks. We only optimized hyperparameters for the Guay benchmark – we added a Gaussian Noise data augmentation – and without a validation dataset this would have run the risk of overfitting to the test set.

– We have added Supplementary file 3 to show cross-validation results for the Guay benchmarks.

3) Given the relatively small size of the CEM500K dataset, how much of a concern is overfitting?

– We agree that overfitting is always a concern. This is exactly why having multiple segmentation benchmarks was essential – indeed we hope that one of the take-aways from our paper is the necessity of testing against multiple benchmarks.

– Our results show clearly that heterogeneity is more important than size, and this mitigates overfitting. Pre-training on the Bloss, 2018 dataset, a larger but less heterogenous image set, led to models that performed well on segmentation tasks drawn from the same organism and tissue (mouse brain) but underperformed when tasked with segmenting data from other contexts (Figure 2E). In the same figure, we showed that, in contrast, models pre-trained on CEM500K performed well across the board.

– That said, we do recognize that there are biases in the current dataset – Figure 1C, C and Appendix—figure 6 show that certain organisms, tissues, and imaging experiments are overrepresented in CEM500K. As more diverse images are added to CEM500K, we expect these disparities will be reduced.

4) Can the authors comment on the resolution of features that are likely relevant in their CEM500K-moco model? What are the prospects of using MoCoV2 to recognize higher-resolution features in cryo-EM images?

– We anticipate that the CEM500K model should be relevant for typical volume EM experiments, where data is acquired at pixel spacing greater than 2 nm (see Figure 1B, and results for Kasthuri++, pixel spacing of 3nm). Relevance will likely taper off at resolutions higher than this. However, it should be possible to finetune the CEM500K pre-trained weights on higher resolution TEM images, following approaches reported for other imaging modalities (Touvron, Vedaldi, Douze, and Jégou, 2019).

– Re. MoCoV2 for cryo EM. For cryotomograms of cells, it is possible that MoCoV2 pre-training can be used for cellular features captured at higher resolutions; of course, this will require a different pre-training dataset, a possible resource in the future. For CryoEM SPA, we think this would be more challenging. The main requirement of MoCoV2 is that images are distinguishable from each other; fields of many near-identical particles but distributed randomly from image to image would therefore present a daunting challenge to MoCoV2 but may find use for detection of rare orientations.

Reviewer #2 (Recommendations for the authors):1) This work clearly targets pixel classification on 2D images, however the 2D focus is not clear in the beginning and only discussed late (Discussion) where 3D approaches are generally dismissed for no other reason than that they are not applicable to 2D images and more expensive. This dismissal is invalid and also unnecessary and should be removed. The 2D focus should become obvious in the Abstract and Introduction of the paper since the term "image" is used for nD data in the community.

Thank you for your positive comments, we have addressed this point at the start of the rebuttal.

2) Introduction, third paragraph: add paragraph.

Thank you, the large paragraph has been split to create a new one.

3) The hashing algorithm used for identifying near duplicates (https://www.hackerfactor.com/blog/index.php?/archives/529-Kind-of-Like-That.html) may not be optimal but this probably doesn't do much harm as its main purpose is to remove nearby sections from 3D volumes.

We agree, hashing is a somewhat coarse method for deduplication, however it is simple and fast. Importantly, the algorithm reduces ~80% of redundant images and visual inspection of the remaining images show satisfactory dissimilarities.

4) Could CEMdedup be useful for tasks that contain empty resin or nuclei?

It is true that in the filtering step we remove images from CEMdedup that exclusively or predominantly consisted of uniform intensity content like resin and the interior of nuclei. However, there are still many images in the final dataset CEM500K that contain patches of nuclei and empty resin, and indeed we show that for a nuclei segmentation task (Perez), models pre-trained on CEMdedup performed similarly to those pre-trained on CEM500K (nuclei IoU of 0.9890 vs. 0.9891 respectively, see source data file). Therefore, we think that the final dataset should suffice for these tasks.

5) Conclusion: Pretraining dataset should include data relevant for the benchmark task. Does unrelated data necessarily make the benchmark performance better? How would CEM500K + Bloss pre-training perform for the mouse data?

– This is a nuanced and excellent point. A key challenge is that “relatedness” of the concepts is some complex (and unknown) function of organism, tissue, sample prep, imaging approach, resolution etc., which is impossible to determine a priori. Therefore in some ways one could think of providing this highly heterogenous image data as hedging our bets, i.e., not to maximize performance on any one benchmark task but rather on the set of all possible cellular EM segmentation tasks. Results comparing CEM500K to Bloss (see Figure 2E) suggest that the improved generality that comes from pre-training on many unrelated datasets does not hinder the model from performing well on specialized downstream tasks such as the mouse brain benchmarks.

– CEM500K includes >15% of brain tissue (see Figure 1C) so the combination of CEM500K and Bloss for pre-training may improve performance on mouse brain data; indeed one possible use-case of CEM500K would be as an add-on to boost performance of specific or homogenous pre-training data sets. We note however that this would heavily skew the pre-training dataset toward a particular biological context, likely decreasing the general utility of resulting models while also increasing computational costs.

6) CEM500K MoCoV2 training included 360° rotations, i.e. the increased rotation invariance compared to ImageNet is by design of the training, not necessarily the data.

Thank you for pointing this out, our wording was ambiguous. Yes, CEM500K-moco’s better invariance to rotation is due to the training augmentations. Pre-training on ImageNet, where object orientation matters, never includes 360° rotations. We have added a note to the section in the manuscript to reflect this point.

7) Subsection “CEM500K models are largely impervious to meaningful image augmentations”: Why would models gain robustness to rare and random events such as artifacts? Weren't such images excluded from the dataset?

Thank you for pointing this out, we were imprecise in our definition of “artifacts”. The artifacts that we filtered out were related to reconstruction artifacts, usually resulting from improper reporting of pixel data. We did NOT filter out experimental artifacts such as those arising from heavy metal stains, FIB milling artifacts, brightness/contrast changes, and we did not exclude misalignments/missing slices from 3-D reconstructions. We relabel the artifact images as “reconstruction artifacts” to be more precise, and we explicitly mention this in the Materials and methods. Since these artifacts occur occasionally and at random with respect to meaningful cellular features, we expect that models will learn to ignore these features.

8) Subsection “CEM500K models learn biologically relevant features”: Could this also be just high contrast/high gradient regions which makes sense for the given training task? I.e. is the focus on objects of interest may be a trivial consequence of objects of interest being the only thing present/distinguishable in the image?

Yes, this is correct. However, sample preparation protocols in EM are designed to specifically accentuate objects of interest (organelles etc.), while other high-contrast features (sample prep artifacts for example) show no consistent patterns. Therefore, the model learns that there are “things” in an image that are distinguishable from each other and that the rest of the image is “background”, we exploit this knowledge for accurate object detection and segmentation of EM images.

9) Subsection “Fully trained CEM500K models achieve state-of-the-art results on EM benchmarks”: The random initialization score for CREMI of 0.0 is surprising. Is this an indication that the training setup or this particular network architecture have other issues that are unrelated to the pre-training? Others have successfully trained synapse detection networks without pre-training.

The 0.0 score results from the shortened training schedule used for pre-trained models. The CREMI Synaptic Cleft benchmark uniquely has a large foreground-background imbalance (very few pixels in the image correspond to the small feature that is the synaptic cleft) and we trained our models using a binary cross entropy loss, which does correct for that imbalance. As a result, randomly initialized models take a much longer time to achieve non-zero results. In the source files released with this manuscript, we include results for 30,000 training iterations for this unique case (compared to the 5,000 reported in the paper).

10) Discussion, second paragraph: This is not correct for CREMI which is data from the large FAFB dataset, and also itself contains more data than necessary in the padded volumes available for download. I do not find it necessary to perform this experiment but it is ok to simply say that this hasn't been done without falsely asserting that it wasn't possible.

Thank you for pointing this out. The lack of heterogeneity of such a dataset is more of a factor than size, and we have updated the manuscript accordingly.

11) I do not understand this statement: "[…] moreover, the elimination of 3D context from volume EM datasets ensures that the shared data can only reasonably be used for DL." Is it necessary?

Our goal with this statement was to assuage researchers who may be reticent to share unpublished data that contributions to CEM500K could only be used for DL work.

12) Materials and methods: The CREMI synaptic cleft evaluation metric is public including code to execute it. Ground truth for test data, however, is secret. Did you submit your results for evaluation? Like above, it is not necessary to do this but the statement "[…] the training and test datasets did not have an official evaluation metric, and the ground truth segmentations were not publicly available[…]" is not correct. May be replace with "The test data for the CREMI Synaptic Cleft benchmark is not publicly available and the challenge uses a different evaluation metric than IoU. Therefore[…]".

Yes, we did submit results which achieved very good F-scores but scored lower on the CREMI metric. As the CREMI leaderboard itself states, “We are not including the F-Score for now, as our current way of computing it can lead to unfair comparison.” So we side-step this point and we replace the text in the manuscript with the suggested wording.

Reviewer #3 (Recommendations for the authors):– Would you please clarify the statement “a model's ability to generalize to unseen images”? I would like to see more clarification regarding this statement as it seems that it ignores the definition of transfer learning problems. Transfer learning techniques are applied where there exists lack of data. Therefore, providing datasets with large numbers of data does not seem to be a good solution for these problems though it is still a valuable dataset for many problems.

In our manuscript, we write that there are two ways by which more robust models can be generated: either a massive amount of annotated data is provided (possible but impractical for a typical biological lab), or transfer learning can be used. Here, by pre-training on existing heterogenous unlabeled data, a model would require far less labeled data to prevent overfitting and perform well on unseen data. We now add a note clarifying this point in the paragraph.

– In the first part of the paper, it is referred to "six publicly available dataset". I know that you name them in the later part of the paper but it is a question for the reader. It would be better to name them in these sections as well.

We have updated the manuscript and now name the benchmarks in the Introduction.

– It would be better if the minimum amount of outperformance is defined when it is presented "significantly outperformed randomly initialized".

Thank you, we have removed the word “significantly” from the sentence.

– Is there any reference/ proof for the following claim: "Although it is currently impossible to determine a priori what data will be useful for a model, we expect that this removal of significant redundancies in the image dataset is unlikely to result in the loss of meaningful information for DL model training."

The determination of usefulness of specific data for a model is a point of intense work in the DL field, but rather than segueing into this discussion, we simply remove the line for brevity. We note however that current state-of-the-art supports our assertion here.

– What would happen if you keep the duplicate images and use MoCoV2 algorithm?

When duplicates are not removed (i.e., the pre-training dataset is CEMraw), we show that performance suffers, we report these results in Figure 2D. It is possible that the main impact of duplicates is to slow down model training as iterations are consumed relearning the same features already seen multiple times previously in the dataset.

– How about not applying the MoCoV2 algorithm and using other algorithms? Have you considered ablation studies in this regard?

Our goal is this work was to prove the usefulness of the CEM500K dataset and demonstrate its application for unsupervised pre-training. To this end we are agnostic to the actual algorithm used (and indeed we consider this an advantage of our approach), making ablation studies unnecessary. We report high performances with our choices, but we have also open-sourced the dataset and our code to encourage the EM and DL communities to investigate other algorithms that may perform better.

– The following statements are unclear and I appreciate it if you clarify them more.1) "These results emphasize the necessity of evaluating deep learning algorithms and pre-training datasets on multiple benchmarks before drawing conclusions about their quality."

By “quality” in this context we mean a model’s biases. If one were to use only a single benchmark from any one organism and tissue (e.g., mouse brain), models showing a good performance may or may not perform well on segmentation tasks from random unrelated EM data – a point that could never be tested unless there were multiple disparate benchmarks. Therefore, multiple benchmarks are needed to determine if a model will be generally useful for an arbitrary EM segmentation task.

2) “We expect that the superior robustness to variations in cellular EM data baked into CEM500K-moco should simplify the process of adjusting to new tasks.”

What we mean here is that the heterogeneity of CEM500K combined with the MoCoV2 algorithm results in the sampling of a wide variety of possible cellular EM features and also robustness to such variations in tasks. To give a concrete example, consider image rotations. When presented with a new EM segmentation task, CEM500K-moco pre-trained models (as opposed to say ImageNet pre-trained models) will not need to learn rotation invariance because, as we demonstrated, they are already largely invariant to image rotation (see Figure 3A). In short, the invariances in CEM500K-moco models put it in a better starting position for adjusting to a new EM image task resulting in faster learning and better performances (Figure 4).